# ROBUST POLICY GRADIENT OPTIMIZATION THROUGH ACTION PARAMETER PERTURBATION IN REINFORCEMENT LEARNING

## ABSTRACT

Policy gradient methods have achieved strong performance in reinforcement learning, yet remain vulnerable to premature convergence and poor generalization, especially in on-policy settings where exploration is limited. Existing solutions typically rely on entropy regularization or action noise, but these approaches either require sensitive hyperparameter tuning or alter the interaction dynamics rather than the optimization process itself. In this paper, we propose Robust Policy Optimization (RPO), a policy gradient method that introduces perturbations to the policy parameters only during optimization. This approach smooths the loss landscape and implicitly regularizes learning, reducing sensitivity to local irregularities while leaving policy behavior during data collection unchanged. We provide a theoretical perspective showing that RPO implicitly biases updates toward flatter and more stable solutions. Empirically, RPO significantly improves upon PPO and entropy-regularized variants across diverse continuous control benchmarks, achieving faster convergence, higher returns, and greater robustness.

## 1 INTRODUCTION

Reinforcement learning (RL) has demonstrated impressive capabilities across a wide range of complex decision-making tasks, including robotic control (Schulman et al., 2017) and game-playing (Mnih et al., 2015; Silver et al., 2018; 2016). Despite these successes, achieving robust and generalizable behavior remains a core challenge, particularly for on-policy algorithms such as Proximal Policy Optimization (PPO) (Schulman et al., 2017). These methods often converge prematurely to overly deterministic policies, exploiting suboptimal strategies that perform well in the short term but fail to generalize across varying conditions.

This tendency is partly due to the local nature of gradient-based policy optimization, which can overfit to short-horizon reward signals and produce policies that generalize poorly under distribution shifts. Such policies often exhibit low entropy and limited adaptability (Mnih et al., 2016; Ahmed et al., 2019; Nikishin et al., 2022), making them brittle in practice. Prior work has shown that encouraging smoother optimization dynamics, for instance through implicit regularization, can improve robustness and generalization in both supervised learning and reinforcement learning.

A common strategy is to introduce entropy regularization (Mnih et al., 2016; Ahmed et al., 2019), adding an entropy bonus to the optimization objective to encourage stochasticity. While effective in some cases, this approach typically requires careful tuning of entropy coefficients and is highly sensitive to hyperparameter selection (Andrychowicz et al., 2020). Other approaches such as action noise injection (Fujimoto et al., 2018; Haarnoja et al., 2018) improve exploration by perturbing actions during environment interaction. However, these methods alter the data collection process rather than the optimization dynamics, often leading to unstable learning or reduced sample efficiency.

We propose **Robust Policy Optimization (RPO)**, a simple and effective alternative that introduces perturbations only during the optimization phase while leaving environment interaction unchanged. By perturbing policy parameters at training time but not during rollouts, RPO smooths the optimization landscape and implicitly regularizes learning. This design steers policy updates away from brittle solutions and toward more stable ones, without requiring entropy tuning or injecting randomness into the interaction process.

**Comparison to Existing Methods.** Our approach differs fundamentally from prior noise-based methods. Off-policy algorithms such as DDPG and TD3 (Lillicrap et al., 2015; Fujimoto et al., 2018) inject noise directly into actions during interaction or into target networks during critic updates, thereby modifying the data distribution. In contrast, RPO leaves environment interaction untouched and applies perturbations only during optimization. Similarly, NoisyNet (Fortunato et al., 2018) introduces learnable noise into network weights to drive exploration, while PGPE shifts randomness to the parameter level before trajectory generation. RPO instead applies explicit and fixed perturbations only during gradient updates, providing a structured form of optimization-time regularization rather than an exploration mechanism. This separation makes RPO complementary to existing approaches while addressing robustness from a fundamentally different angle.

**Theoretical Perspective.** From a theoretical standpoint, RPO can be interpreted as optimizing a smoothed version of the policy objective, obtained by averaging over a neighborhood of perturbed policies. This smoothing effect implicitly biases learning toward flatter regions of the optimization landscape, which are associated with improved stability and generalization. While the formal derivation assumes perturbations of the full parameter vector, in practice RPO perturbs only the action-related parameters, and the same reasoning applies within this subspace (see Appendix C.2 for details).

We evaluate RPO across continuous control tasks from DeepMind Control (Tunyasuvunakool et al., 2020), OpenAI Gym (Brockman et al., 2016), PyBullet (Coumans & Bai, 2016–2021), and Nvidia IsaacGym (Makoviychuk et al., 2021). Across these diverse benchmarks, RPO consistently outperforms PPO and entropy-regularized PPO, demonstrating faster convergence, higher asymptotic returns, improved sample efficiency, and greater robustness. RPO outperforms or matches PPO in 93% of environments tested, for a total of 57 out of 61 environments. On the DeepMind Control Suite in particular, RPO achieves more than twice the improvement over PPO in terms of overall performance.

**Contributions.**

- We introduce **Robust Policy Optimization (RPO)**, a novel policy gradient method that applies optimization-time perturbations to the policy parameters. This perturbation acts as a form of loss smoothing, encouraging robust and generalizable policy learning without modifying environment interaction.
- We provide a **theoretical perspective** showing that optimization-time perturbations implicitly regularize policy optimization by biasing updates toward flatter, more stable solutions.
- We conduct extensive experiments on a range of **continuous control tasks**, demonstrating that RPO consistently outperforms strong on-policy baselines in terms of sample efficiency, robustness, and stability.

## 2 PRELIMINARIES: POLICY GRADIENT

We consider a standard Markov Decision Process (MDP) defined by the tuple $(\mathcal{S}, \mathcal{A}, P, r, \gamma)$, where $\mathcal{S}$ is the state space, $\mathcal{A}$ is the action space, $P(s' \mid s, a)$ is the transition probability function, $r(s, a)$ is the reward function, and $\gamma \in (0, 1)$ is the discount factor. The agent's objective is to learn a policy $\pi(a \mid s; \theta)$, parameterized by $\theta \in \mathbb{R}^d$, that maximizes the expected cumulative return:

$$J(\theta) = \mathbb{E}_{\tau \sim \pi_\theta} \left[ \sum_{t=0}^{\infty} \gamma^t r(s_t, a_t) \right], \tag{1}$$

where $\tau = (s_0, a_0, s_1, a_1, \dots)$ denotes a trajectory generated by following policy $\pi_\theta$.

Policy gradient methods directly optimize $J(\theta)$ by estimating its gradient using the policy gradient theorem (Sutton et al., 1999):

$$\nabla_\theta J(\theta) = \mathbb{E}_{\pi_\theta} \left[ \nabla_\theta \log \pi_\theta(a \mid s) A^{\pi_\theta}(s, a) \right], \tag{2}$$

where $A^{\pi_\theta}(s, a) = Q^{\pi_\theta}(s, a) - V^{\pi_\theta}(s)$ is the advantage function, representing how much better an action is compared to the expected value of the current policy in state $s$. In practice, the advantage is

estimated using temporal-difference methods or generalized advantage estimation (GAE) (Schulman et al., 2016), which balances bias and variance by incorporating multi-step value targets.

This advantage-weighted objective enables actor-critic methods to scale to complex environments by providing lower-variance gradient estimates. To further stabilize training (Kakade, 2001), Trust Region Policy Optimization (TRPO) (Schulman et al., 2015) and Proximal Policy Optimization (PPO) (Schulman et al., 2017) constrain the size of policy updates using trust region constraints or clipped surrogate objectives. Nonetheless, these methods can still suffer from premature convergence to low-entropy or brittle policies (Andrychowicz et al., 2020; Nikishin et al., 2022).

## 3 RELATED WORK

**Policy Gradient.** Since the inception of practical policy optimization methods like PPO, TRPO, and Natural Policy Gradient, several studies have investigated different algorithmic components (Engstrom et al., 2019; Andrychowicz et al., 2020; Ahmed et al., 2019; Ilyas et al., 2019). Entropy regularization has been widely used to encourage exploration and improve optimization (Williams & Peng, 1991; Mei et al., 2020; Mnih et al., 2016; Ahmed et al., 2019). However, setting an appropriate entropy coefficient can be difficult in practice, and empirical results suggest that its benefits are often environment-dependent (Andrychowicz et al., 2020). Our work instead pursues a different direction: rather than injecting entropy into the interaction process, we introduce perturbations directly during optimization, thereby regularizing the policy gradient updates without modifying the collected trajectories.

**Noise-Injected Policies.** Adding noise to actions is a standard technique for improving exploration in reinforcement learning. Deterministic policy methods such as DDPG (Lillicrap et al., 2015) and TD3 (Fujimoto et al., 2018) employ action noise during environment interaction, thereby directly altering the distribution of collected trajectories. This differs fundamentally from our approach, which applies perturbations only during the policy update phase and leaves the environment interaction untouched. While TD3 also employs perturbations during learning by injecting noise into target actions, it is an off-policy actor–critic method relying on target networks. In contrast, our method is developed in the context of on-policy policy gradient algorithms such as PPO, where perturbations act on the policy parameters to smooth the optimization landscape.

**Parameter-Space Noise.** An alternative to action-level noise is parameter-space noise, where perturbations are applied to the policy parameters themselves. Plappert et al. (Plappert et al., 2018) showed that parameter-space noise can lead to more consistent exploration across states. Similarly, NoisyNet (Fortunato et al., 2018) perturbs network weights with learned noise parameters to drive exploratory behavior. Our method shares the idea of parameter perturbation but differs in both mechanism and purpose. First, unlike NoisyNet, which learns noise parameters as part of the model and affects the policy used for environment interaction, our perturbations are fixed, explicitly applied, and only affect the optimization dynamics. Second, whereas Plappert et al. (Plappert et al., 2018) proposed perturbations for exploration during trajectory generation, we apply perturbations only during gradient estimation, which regularizes optimization rather than exploration. This distinction allows for a clear theoretical characterization of our method as introducing a curvature-dependent regularizer.

**Policy Gradients with Parameter-Based Exploration (PGPE).** Another related direction is PGPE (Sehnke et al., 2010), which shifts randomness from the action level to the parameter level by perturbing policy parameters prior to environment interaction. This eliminates per-step action noise and is often studied with deterministic policies. In contrast, our method does not alter the trajectory generation process. Instead, perturbations are applied exclusively during the gradient computation step, separating optimization regularization from exploration. This difference makes RPO complementary to PGPE and related methods (Montenegro et al., 2024; Zhao et al., 2011; Metelli et al., 2018; 2020; Xu et al., 2020; Likmeta et al., 2020): whereas PGPE modifies exploration at the trajectory level, RPO stabilizes optimization dynamics without changing the data-collection process.

**Data Augmentation.** Another avenue for increasing policy entropy is through observation-level data augmentation. Methods such as RAD and DrAC (Laskin et al., 2020; Raileanu et al., 2020) show that random transformations of input observations can regularize learning and improve generalization

in visual RL tasks. However, augmentation often requires environment-specific design choices and domain knowledge to be effective. In contrast, our method is domain-agnostic: perturbations are applied in parameter space during optimization, with no assumptions about the environment modality.

**Policy Optimization.** Numerous implementation refinements have contributed to the empirical success of modern policy optimization, including Generalized Advantage Estimation (Schulman et al., 2016), normalization of advantages (Andrychowicz et al., 2020), and clipped policy and value objectives (Schulman et al., 2017; Engstrom et al., 2019; Andrychowicz et al., 2020). These improvements are widely adopted in standard implementations (Huang et al., 2022a;b). Our method is complementary to such techniques: we adopt them in our implementation for a fair comparison and show that optimization-time perturbations provide additional robustness and performance benefits on top of these established practices.

## 4 METHODOLOGY: ROBUST POLICY OPTIMIZATION (RPO)

### 4.1 ALGORITHMIC DESCRIPTION

Robust Policy Optimization (RPO) is a modified policy gradient method that introduces perturbations to the policy parameters during the optimization phase, while keeping the data collection policy unchanged. This separation aims to ensure the exploration behavior remains stable during environment interaction, while optimization benefits from smoother gradient estimates. Unlike traditional entropy regularization, which requires manually tuned coefficients and modifies the reward objective, RPO regularizes implicitly by averaging gradient updates over a neighborhood in parameter space. The details of the RPO method are presented in Algorithm 1.

---

**Algorithm 1** Robust Policy Optimization (RPO)

---

1: **Initialize:** Policy parameters $\theta$, experience buffer $\mathcal{D}$
2: **for** each iteration **do**
3: $\quad \mathcal{D} \leftarrow \emptyset$ $\hfill \triangleright$ Initialize batch storage
4: $\quad$ **for** each environment step $t$ **do**
5: $\quad\quad (\mu_t, \sigma_t) \leftarrow \pi_\theta(s_t)$ $\hfill \triangleright$ Compute action distribution
6: $\quad\quad a_t \sim \mathcal{N}(\mu_t, \sigma_t)$ $\hfill \triangleright$ Sample action
7: $\quad\quad$ Execute $a_t$, observe $s_{t+1}, r_t$
8: $\quad\quad$ Store transition $(s_t, a_t, r_t, s_{t+1})$ in $\mathcal{D}$
9: $\quad$ **end for**
10: $\quad$ Compute advantages $A_t$ for each $(s_t, a_t) \in \mathcal{D}$
11: $\quad$ **for** each transition $(s_t, a_t, A_t) \in \mathcal{D}$ **do**
12: $\quad\quad (\mu_t, \sigma_t) \leftarrow \pi_\theta(s_t)$ $\hfill \triangleright$ Compute policy mean and std
13: $\quad\quad$ Sample perturbation $z \sim \mathcal{U}(-\alpha, \alpha)$
14: $\quad\quad$ Set perturbed mean $\mu'_t = \mu_t + z$
15: $\quad\quad$ Compute log-probability $\log \pi_\theta(a_t \mid s_t; \mu'_t, \sigma_t)$
16: $\quad\quad g \leftarrow g + \nabla_\theta \log \pi_\theta(a_t \mid s_t; \mu'_t, \sigma_t) A_t$ $\hfill \triangleright$ Accumulate policy gradient
17: $\quad$ **end for**
18: $\quad$ Update $\theta$ using accumulated gradients
19: **end for**

---

### 4.2 THEORETICAL ANALYSIS

The key intuition behind Robust Policy Optimization (RPO) is that perturbing the policy parameters during the optimization phase introduces a controlled stochasticity. By decoupling perturbation from data collection, RPO aims to avoid the instability associated with action-space noise, while still encouraging diversity in gradient updates. Unlike traditional entropy-regularized methods, which require manual tuning of entropy coefficients and modify the reward function, RPO implicitly maintains policy stochasticity through optimization-time noise, without altering the objective.

In the continuous control setting, the policy is typically modeled as a Gaussian distribution:

$$\pi_\theta(a \mid s) = \mathcal{N}(\mu_\theta(s), \sigma_\theta(s)), \tag{3}$$

where $\mu_\theta(s)$ and $\sigma_\theta(s)$ are the outputs of a neural network parameterized by $\theta$. Gaussian policies are widely used in continuous control due to their analytical tractability and compatibility with reparameterization-based gradient estimation.

During optimization, RPO perturbs the mean as:

$$\mu' = \mu + z, \quad z \sim \mathcal{U}(-\alpha, \alpha), \tag{4}$$

resulting in a modified policy distribution:

$$\tilde{\pi}_\theta(a \mid s) = \mathbb{E}_z \left[ \mathcal{N}(a \mid \mu + z, \sigma) \right]. \tag{5}$$

This formulation ensures that policy gradient updates are influenced by a neighborhood of parameter values rather than a single deterministic policy. As a result, RPO naturally prevents premature convergence to deterministic, low-entropy solutions and encourages robust, diverse behavior.

We next provide a formal analysis of how this perturbation affects gradient estimation and induces an implicit regularization effect that improves generalization and optimization stability.

### 4.3 UNBIASEDNESS OF THE GRADIENT ESTIMATOR

A desirable property of any modification to the policy gradient method is that it does not introduce bias in the gradient estimate. In this subsection, we show that RPO preserves unbiasedness despite applying perturbations to the policy parameters during optimization.

The standard policy gradient is:

$$\nabla_\theta J(\theta) = \mathbb{E}_{s \sim \rho^\pi, a \sim \pi_\theta} \left[ \nabla_\theta \log \pi_\theta(a \mid s) A(s, a) \right]. \tag{6}$$

In RPO, the policy distribution is perturbed during optimization:

$$\pi_\theta^{\text{pert}}(a \mid s; z) = \mathcal{N}(\mu_\theta(s) + z, \sigma_\theta(s)), \quad z \sim \mathcal{U}(-\alpha, \alpha), \tag{7}$$

but the data used for gradient estimation (states and actions) is collected using the unperturbed policy $\pi_\theta$.

The RPO update computes:

$$\tilde{g}(\theta) = \mathbb{E}_z \left[ \nabla_\theta \log \pi_\theta(a \mid s; \mu + z, \sigma) A(s, a) \right]. \tag{8}$$

Because the perturbation $z$ is sampled independently from a symmetric zero-mean distribution and does not affect the data distribution, we can apply linearity of expectation:

$$\mathbb{E}_z \left[ \nabla_\theta \log \pi_\theta(a \mid s; \mu + z, \sigma) \right] = \nabla_\theta \mathbb{E}_z \left[ \log \pi_\theta(a \mid s; \mu + z, \sigma) \right]. \tag{9}$$

Since the policy is Gaussian and smooth, this expectation can be viewed as convolution with a uniform kernel, which preserves the mean gradient direction. Hence:

$$\mathbb{E}_z \left[ \nabla_\theta \log \pi_\theta(a \mid s; \mu + z, \sigma) \right] = \nabla_\theta \log \pi_\theta(a \mid s), \tag{10}$$

and the full RPO estimator becomes:

$$\mathbb{E}_z[\tilde{g}(\theta)] = \nabla_\theta \log \pi_\theta(a \mid s) A(s, a). \tag{11}$$

Thus, RPO does not introduce any bias into the policy gradient estimator (Sutton et al., 1999). It only modifies the way the gradient is computed during the optimization step, without affecting the trajectory distribution or introducing systematic drift. This unbiasedness is crucial for ensuring that RPO retains the convergence guarantees of standard policy gradient methods (Sutton et al., 1999; Schulman et al., 2015; Kakade, 2001), while benefiting from the regularization and smoothing effects of the perturbation.

## 4.4 Induced Regularization Effect

Beyond preserving the unbiasedness of the policy gradient, RPO also induces an implicit regularization effect. By introducing perturbations to the policy parameters during optimization, the loss landscape becomes smoothed, discouraging convergence to narrow optima and improving generalization.

To illustrate this effect, consider the expected log-likelihood under perturbation:

$$\mathbb{E}_z\left[-\log \pi_\theta(a \mid s; \mu + z, \sigma)\right]. \tag{12}$$

Due to the convexity of the negative log-likelihood in the perturbed mean and the symmetry of the perturbation distribution, this expectation introduces a regularization term. Specifically, using a second-order Taylor expansion, it can be shown (details in Appendix) that:

$$\mathbb{E}_z\left[-\log \pi_\theta(a \mid s; \mu + z, \sigma)\right] \approx -\log \pi_\theta(a \mid s) + \frac{\alpha^2}{6\sigma^2}. \tag{13}$$

This extra term penalizes low variance in the action distribution by discouraging overly sharp likelihood peaks. As a result, RPO acts similarly to entropy regularization, but without requiring explicit entropy bonuses or tuning of exploration coefficients. Instead, it maintains stochasticity through parameter-space smoothing, providing a self-regularizing mechanism that encourages more robust, generalizable policies. We provide a full derivation of this regularization effect in Appendix.

## 4.5 Loss Landscape Smoothing in RPO

To better understand the effect of optimization-time perturbations, we analyze how RPO modifies the policy gradient objective. Our analysis shows that perturbing parameters during training is equivalent to optimizing a smoothed version of the loss, which can be interpreted as convolving the original objective with a uniform kernel. A second-order expansion reveals that this smoothing introduces an implicit regularization term proportional to the trace of the Hessian, biasing updates toward flatter regions of the landscape that are associated with improved stability and generalization. While the derivation is presented for the full parameter vector, in practice RPO perturbs only the action-related parameters (e.g., the mean of the Gaussian policy), and the same reasoning applies within this restricted subspace. We provide the full derivation and further discussion in Appendix C.2.

## 5 Experiments

### 5.1 Setup

**Environments** We conducted experiments on continuous control task from four reinforcement learning benchmarks: DeepMind Control (Tunyasuvunakool et al., 2020), OpenAI Gym Mujoco control tasks (Brockman et al., 2016), PyBullet (Coumans & Bai, 2016–2021), and Nvidia IsaacGym (Makoviychuk et al., 2021). These benchmarks contain diverse environments with many different tasks, from low to high-dimensional environments (observations and actions space). Thus our evaluation contains a diverse set of tasks with various difficulties.

**Baselines** We compare our method RPO with the standard **PPO** (Schulman et al., 2017) algorithm. Here our method RPO uses the perturbed Gaussian distribution to represent the action output from the policy network. In contrast, the PPO uses standard Gaussian distribution to represent its action output. Unless otherwise mentioned, we tested on Gaussian distribution. However, we provide separate experiments for different parameterized distributions, including Laplace and Gumbel.

Another common approach to increase entropy is to use the **Entropy Regularization** (Mnih et al., 2016) in the RL objective. A coefficient determines how much weight the policy would give to the entropy. We observe that various weighting might result in different levels of entropy increment. We use the entropy coefficient 0.0, 0.01, 0.05, 0.5, 1.0, and 10.0 and compare their performance in entropy and, in return, with our RPO algorithm. Note that our method does not use the entropy coefficient hyper-parameters.

To account for the stochasticity in the environment and policy, we run each experiment several times and report the mean and standard deviation. Unless otherwise specified, we run each experiment with **10 random seeds**. Further implementation details are in the Appendix.

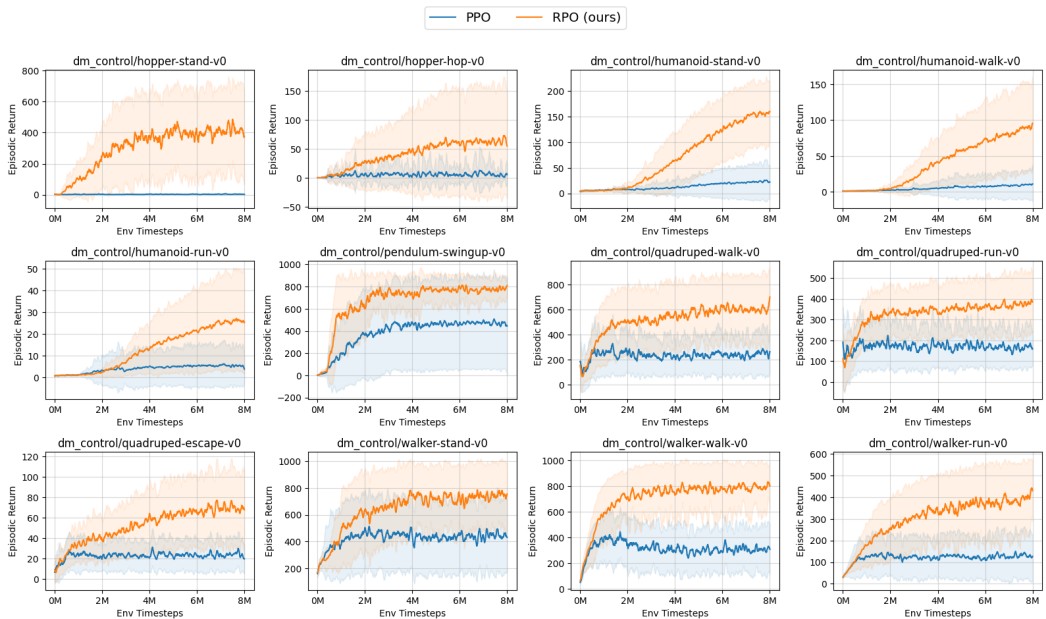

Figure 1: Results on DeepMind Control Environments. PPO agent fails to learn any useful behavior and thus results in low episodic return in some environments (humanoid: stand, walk, and run and hopper: stand, and hop). Overall, our method RPO performs better in episodic return. In many settings, PPO agents stop improving their performance after around 2M timestep while RPO consistently improves over the entire training time.

## 5.2 COMPARISON WITH PPO

We notice a consistent improvement in the performance of our method in many continuous control environments compared to standard PPO. This evaluation is the direct form of comparison where no method uses any aid (such as data augmentation or entropy regularization) in the entropy value. Figure 1 shows results comparison on *DeepMind Control Environments*. Results on more environments and entropy comparisons are in the Appendix.

In most scenarios, our method RPO shows consistent performance improvement in these environments compared to the PPO. In some environments, such as humanoid stand, humanoid run, and hopper hop, the PPO agent fails to learn any useful behavior and thus results in low episodic return. In contrast, our method RPO shows better performance and achieves a much higher episodic return. The RPO also shows a better mean return than the PPO in other environments. In particular, RPO achieves equal or better performance

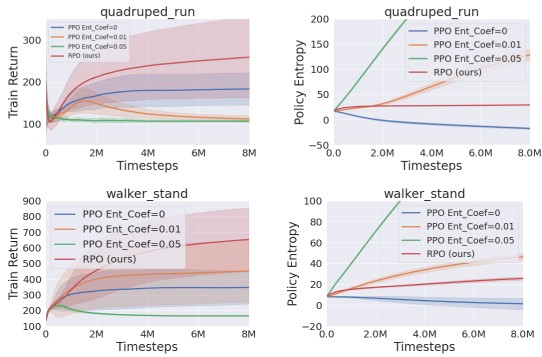

Figure 2: Comparison with Entropy Regularization. While entropy regularization can help with careful tuning (e.g., walker stand), poor coefficient choices often degrade performance. In contrast, RPO consistently outperforms PPO and all entropy-tuned baselines.

than PPO in 93% of the environments tested, covering all 48 tasks from the DeepMind Control Suite, one task from OpenAI Gym, one task from Nvidia IsaacGym and 7 out of 11 MuJoCo v4 tasks, for a total of 57 out of 61 environments. On the DeepMind Control Suite in particular, RPO delivers more than a twofold improvement over PPO in overall performance. Together, these findings establish

RPO as a practical and reliable drop-in replacement for PPO, offering enhanced stability, sample efficiency, and generalization without additional tuning.

In many settings, such as in quadruped (walk, run, and escape), walker (stand, walk, run), fish swim, acrobot swingup, the PPO agents stop improving their performance after around 2M timesteps. In contrast, our agent RPO shows consistent improvement over the course of training. This performance gain might be due to the proper management of policy entropy, as in our setup, the agent is encouraged to keep exploring as the training progresses. On the other hand, the PPO agent might settle in sub-optimal performance as the policy entropy, in this case, decreases as the agent trains for more timesteps. These results show the effectiveness of our method in diverse control tasks with varying complexity. *Experimental results for other continuous benchmarks, such as those in IsaacGym, Gym(nasium) Mujoco control, and PyBullet environments, are provided in the Appendix.*

## 5.3 Comparison with Entropy Regularization

A way to control entropy is to use an entropy regularizer (Mnih et al., 2016; Ahmed et al., 2019), which often shows beneficial effects. However, it has been observed that increasing entropy in such a way has little effect in specific environments (Andrychowicz et al., 2020). These results show difficulty in setting proper entropy throughout the agent's training.

Due to the variety in the environments, the entropy requirement might be different. Thus, improper setting of the entropy coefficient might result in bad training performance. In the tested environments, we observe that sometimes the performance improves with a proper setup of tuned coefficient value and often worsens the performance (in Figure 2). However, our method RPO consistently performs better than PPO and the different entropy coefficient baselines. Importantly, our method

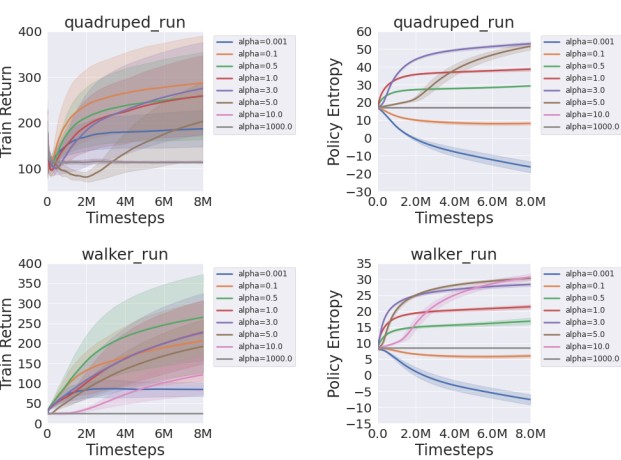

Figure 3: Ablation on $alpha$ values of the uniform distribution for RPO. An $\alpha$ value between $0.1$ to $3$ often results in better performance, while a large value often results in worse performance. The suggested default value is $\alpha = 0.5$.

does not use these coefficient hyper-parameters and still consistently performs better in various environments, which shows our method's robustness in various environment variability.

We observe that in some environments, the coefficient $0.01$ improves the performance of standard PPO with coefficient $0.0$ while increasing the entropy. However, an increase in entropy to $0.05$ and above results in an unbounded entropy increase. Thus, the performance worsens in most scenarios where the agent fails to achieve a reasonable return. Results with all the coefficient variants in the Appendix. Overall, our method RPO achieves better performance in the evaluated environments in our setup. Moreover, our method does not use the entropy coefficient hyperparameter and controls the entropy level automatically in each environment.

## 6 Ablation Study

### 6.1 Effect of $\alpha$ on RPO

We conducted experiments on the $\alpha$ value ranges in the Uniform distribution. Figure 3 shows the return and policy comparison. We observe that the value of $\alpha$ affects the policy entropy and, thus, return performance. A smaller value of $\alpha$ (e.g., $0.001$) seems to behave similarly to PPO, where policy entropy decreases over time, thus hampered performance. Higher entropy values,

such as 1000.0, make the policy somewhat random as the uniform distribution dominates over the Gaussian distribution. This scenario keeps the entropy somewhat at a constant level; thus, the performance is hampered. Overall, a value between 0.1 to 3 often results in better performance.

Due to overall performance advantage, in this paper, we report results with $\alpha = 0.5$ for all environments. We implement a version of our RPO algorithm that varies the $\alpha$ parameter, which controls the amount of uniform perturbation applied to the policy. Specifically, we anneal $\alpha$ from its default value (0.5 in our main experiments) down to zero over the course of training.

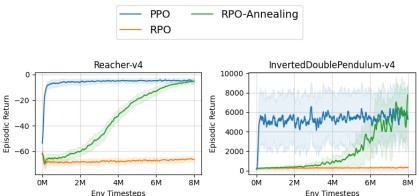

## 6.2 RESULTS ON RPO-$\alpha$ ANNEALING

The value decreases uniformly at each policy update, ensuring that the injected randomness is gradually reduced and ultimately removed by the end of training.

Figure 4: RPO-Annealing on Mujoco-v4. In two challenging cases where the default RPO ($\alpha = 0.5$) fails, the annealed version recovers and eventually matches PPO's performance.

We evaluate RPO-Annealing on challenging Mujoco-v4 environments, where the default RPO setting ($\alpha = 0.5$) fails to learn. In these cases, the annealed version recovers and eventually matches the performance of the base PPO. Notably, RPO can also succeed in these environments by fine-tuning the $\alpha$ value, as shown in Figure 4. Overall, annealing $\alpha$ is a promising strategy that can improve performance and enhance stability. Moreover, proper tuning of $\alpha$ can lead to even better outcomes compared to the default, as evidenced in Figure 4. Further results are in the Appendix.

## 6.3 EFFECT OF ACTION DISTRIBUTION

Figure 5 shows the performance of RPO across different action distributions (Gaussian, Laplace, and Gumbel) on Mujoco and Bullet environments. Empirically, RPO improves performance over PPO even when PPO uses alternative action distributions, demonstrating RPO's robustness. While RPO consistently enhances learning across all tested distributions, we observe that the Gaussian distribution tends to perform best for continuous control tasks. Additional results are provided in the Appendix.

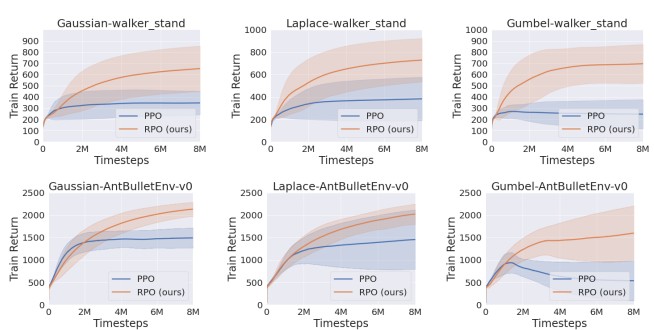

Figure 5: Results on different action distributions. Our method RPO shows improvement compared to base distributions.

## 7 CONCLUSION

We introduced Robust Policy Optimization (RPO), an effective enhancement to policy gradient methods that improves robustness by applying parameter perturbations during optimization. Unlike approaches that rely on entropy regularization or action noise, RPO preserves policy behavior during environment interaction and removes the need for tuning entropy coefficients. Our theoretical perspective shows that RPO implicitly smooths the optimization landscape by discouraging sharp curvature, thereby guiding updates toward flatter and more stable solutions. Empirically, RPO achieves equal or better performance than PPO in 93% of tested environments. These results highlight RPO as a practical and theoretically grounded framework that delivers faster convergence, higher returns, and improved robustness across diverse continuous control benchmarks. Overall, RPO provides a simple yet powerful drop-in replacement for PPO, enabling more consistent and adaptable on-policy learning in reinforcement learning.

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

## A    DERIVATION OF GRADIENT UNBIASEDNESS UNDER PARAMETER PERTURBATION

We now provide a more detailed justification for the identity used in the unbiasedness proof:

$$\mathbb{E}_z \left[ \nabla_\theta \log \pi_\theta(a \mid s; \mu + z, \sigma) \right] = \nabla_\theta \log \pi_\theta(a \mid s). \tag{14}$$

Let $\pi_\theta(a \mid s) = \mathcal{N}(\mu_\theta(s), \sigma_\theta(s))$ be a Gaussian policy. During RPO optimization, the mean is perturbed as $\mu' = \mu + z$, where $z \sim \mathcal{U}(-\alpha, \alpha)$ is independent of the policy parameters $\theta$.

Define the perturbed log-likelihood:

$$\ell_z(\theta) := \log \pi_\theta(a \mid s; \mu + z, \sigma). \tag{15}$$

We want to show:

$$\mathbb{E}_z[\nabla_\theta \ell_z(\theta)] = \nabla_\theta \log \pi_\theta(a \mid s). \tag{16}$$

This is justified using the following steps:

**Step 1: Expand the log-likelihood of the Gaussian.** Let us write:

$$\pi_\theta(a \mid s; \mu + z, \sigma) = \frac{1}{\sqrt{2\pi\sigma^2}} \exp\left(-\frac{(a - \mu - z)^2}{2\sigma^2}\right). \tag{17}$$

Taking the log:

$$\log \pi_\theta(a \mid s; \mu + z, \sigma) = -\frac{(a - \mu - z)^2}{2\sigma^2} - \log(\sqrt{2\pi\sigma^2}). \tag{18}$$

**Step 2: Compute the gradient w.r.t. $\theta$.** We compute $\nabla_\theta \log \pi_\theta(a \mid s; \mu + z, \sigma)$. Since $z$ is independent of $\theta$, and $\mu = \mu_\theta(s)$, the derivative only applies to $\mu$ (and possibly $\sigma$):

$$\nabla_\theta \log \pi_\theta(a \mid s; \mu + z, \sigma) = \frac{(a - \mu - z)}{\sigma^2} \nabla_\theta \mu_\theta(s). \tag{19}$$

**Step 3: Take expectation over $z$.** Now take expectation:

$$\mathbb{E}_z \left[\nabla_\theta \log \pi_\theta(a \mid s; \mu + z, \sigma)\right] = \nabla_\theta \mu_\theta(s) \cdot \frac{1}{\sigma^2} \mathbb{E}_z \left[(a - \mu - z)\right]. \tag{20}$$

Because $z$ is zero-mean and independent of $a$ and $\mu$:

$$\mathbb{E}_z \left[a - \mu - z\right] = a - \mu. \tag{21}$$

So:

$$\mathbb{E}_z \left[\nabla_\theta \log \pi_\theta(a \mid s; \mu + z, \sigma)\right] = \frac{(a - \mu)}{\sigma^2} \nabla_\theta \mu_\theta(s) = \nabla_\theta \log \pi_\theta(a \mid s). \tag{22}$$

Thus, the expectation of the perturbed gradient matches the original unperturbed policy gradient:

$$\mathbb{E}_z \left[\nabla_\theta \log \pi_\theta(a \mid s; \mu + z, \sigma)\right] = \nabla_\theta \log \pi_\theta(a \mid s), \tag{23}$$

which proves that the RPO gradient estimator is unbiased.

# B DERIVATION OF THE REGULARIZATION EFFECT

We derive how Robust Policy Optimization (RPO) induces an implicit regularization effect by applying noise to the policy mean during the optimization phase. Specifically, we show that the expected log-likelihood under perturbation introduces a curvature-dependent penalty term, which prevents variance collapse and promotes higher-entropy, more robust policies.

## B.1 SETUP

Let the policy be Gaussian:

$$\pi_\theta(a \mid s) = \mathcal{N}(\mu_\theta(s), \sigma_\theta(s)), \tag{24}$$

with fixed standard deviation $\sigma = \sigma_\theta(s)$, and $\mu = \mu_\theta(s)$ denoting the mean.

In RPO, we perturb the mean by adding a noise vector $z \sim \mathcal{U}(-\alpha, \alpha)$, leading to:

$$\mu' = \mu + z. \tag{25}$$

We analyze the effect of computing the expected negative log-likelihood over this perturbation:

$$\mathbb{E}_z \left[-\log \pi_\theta(a \mid s; \mu + z, \sigma)\right]. \tag{26}$$

## B.2 NEGATIVE LOG-LIKELIHOOD

Recall the negative log-likelihood for a Gaussian:

$$-\log \pi_\theta(a \mid s; \mu + z, \sigma) = \frac{(a - \mu - z)^2}{2\sigma^2} + \frac{1}{2} \log(2\pi\sigma^2). \tag{27}$$

We expand the squared term:

$$(a - \mu - z)^2 = (a - \mu)^2 - 2z(a - \mu) + z^2. \tag{28}$$

Taking expectation over the uniform distribution $z \sim \mathcal{U}(-\alpha, \alpha)$: - $\mathbb{E}[z] = 0$, - $\mathbb{E}[z^2] = \frac{\alpha^2}{3}$.

Thus:

$$\mathbb{E}_z\left[(a - \mu - z)^2\right] = (a - \mu)^2 + \frac{\alpha^2}{3}. \tag{29}$$

Substituting into the expectation of the log-likelihood:

$$\mathbb{E}_z\left[-\log \pi_\theta(a \mid s; \mu + z, \sigma)\right] = \frac{(a - \mu)^2}{2\sigma^2} + \frac{\alpha^2}{6\sigma^2} + \frac{1}{2}\log(2\pi\sigma^2). \tag{30}$$

The first and third terms correspond to the unperturbed negative log-likelihood. The second term is the added regularization:

$$\frac{\alpha^2}{6\sigma^2}. \tag{31}$$

### B.3 INTERPRETATION

We conclude that:

$$\mathbb{E}_z\left[-\log \pi_\theta(a \mid s; \mu + z, \sigma)\right] \approx -\log \pi_\theta(a \mid s) + \frac{\alpha^2}{6\sigma^2}. \tag{32}$$

This extra term penalizes small standard deviation $\sigma$, thereby preventing entropy collapse and implicitly encouraging broader, smoother action distributions. Importantly, this effect arises without requiring an explicit entropy bonus in the objective function.

Hence, RPO behaves like an entropy-regularized policy gradient method, but achieves this through optimization-time parameter perturbation, without modifying the reward or introducing additional hyperparameters.

## C  LOSS LANDSCAPE ANALYSIS

For clarity of exposition, the following analysis is written assuming perturbations are applied to the entire parameter vector $\theta$. In practice, RPO perturbs only the action-related parameters (e.g., the mean of the Gaussian policy). The same reasoning still applies in this restricted setting: the smoothing effect operates on the subspace of perturbed parameters, and the resulting regularization term involves the corresponding block of the Hessian.

We denote the policy optimization objective as $J(\theta) = \mathbb{E}_{\pi_\theta}\left[\sum_t \gamma^t r_t\right]$, where $\pi_\theta$ is the policy parameterized by $\theta$. In the derivation below, we write $L(\theta) = -J(\theta)$ to emphasize the minimization perspective: smoothing $L(\theta)$ is equivalent to regularizing the maximization of $J(\theta)$. Thus, the curvature-penalization effect derived for $L(\theta)$ directly characterizes how RPO biases policy optimization toward flatter, more robust solutions in terms of $J(\theta)$.

### C.1  IMPLICIT REGULARIZATION THROUGH OPTIMIZATION-TIME PERTURBATION

Recent work in deep learning has shown that flatter regions of the loss landscape are associated with improved generalization (Keskar et al., 2017). This insight has motivated the use of implicit regularization techniques such as dropout (Srivastava et al., 2014), data augmentation, batch normalization in supervised (Ioffe & Szegedy, 2015) and off-policy reinforcement learning (Bhatt et al., 2024). These methods introduce stochasticity during training, which helps models converge to broader optima that generalize better to unseen data.

In reinforcement learning setting issues of generalization and stability arise due to the non-stationary, high-variance nature of policy updates. We are particularly interested in the effect of introducing noise into the policy *parameters during optimization*—not during data collection or interaction with

the environment. This idea can be understood through the lens of optimization-time smoothing, a general technique that modifies the training loss by averaging it over a neighborhood in parameter space.

Specifically, we define a smoothed training objective of the form:

$$\tilde{L}(\theta) = \mathbb{E}_{z \sim \mathcal{U}(-\alpha, \alpha)} \left[ L(\theta + z) \right], \tag{33}$$

where $z \in \mathbb{R}^d$ is a uniformly sampled perturbation vector and $\alpha > 0$ controls the magnitude of the perturbation. This formulation corresponds to a convolution of the original loss with a uniform kernel, resulting in a smoothed version of the loss landscape.

Although no explicit regularization term is added to the objective, this smoothing induces an implicit bias toward flatter solutions. Prior work (Dziugaite & Roy, 2018; Foret et al., 2021) has shown that this effect can be understood through a second-order Taylor expansion:

$$\tilde{L}(\theta) \approx L(\theta) + \frac{\alpha^2}{6} \text{Tr}(HL(\theta)), \tag{34}$$

where $HL(\theta)$ denotes the Hessian of the loss with respect to $\theta$. The trace term penalizes curvature, discouraging convergence to sharp local minima and instead favoring broader, more stable regions of the loss surface.

This theoretical connection between curvature and generalization has been extensively validated in supervised learning (Dziugaite & Roy, 2018; Foret et al., 2021). Sharp minima lead to high sensitivity to small changes in model parameters and poor robustness under distribution shifts. Implicit regularization via loss smoothing helps avoid such solutions, yielding models that are more robust and generalizable.

In the following section, we apply this principle to policy gradient methods by introducing optimization-time parameter perturbations. Our approach leverages this implicit regularization effect to stabilize training and improve generalization in reinforcement learning.

### C.2 LOSS LANDSCAPE SMOOTHING VIA TAYLOR EXPANSION

We show how Robust Policy Optimization (RPO) smooths the policy gradient loss landscape by averaging over perturbed parameters. This leads to a regularization effect that penalizes sharp curvature and encourages more stable, generalizable solutions.

### C.3 INTERPRETATION AS A CONVOLUTION WITH A UNIFORM KERNEL

The RPO objective can be interpreted as a convolution between the loss function and a uniform smoothing kernel. Specifically, the smoothed loss is defined as:

$$\tilde{L}(\theta) = \mathbb{E}_{z \sim \mathcal{U}(-\alpha, \alpha)} \left[ L(\theta + z) \right] = \int_{\mathbb{R}^d} L(\theta + z) p(z) \, dz. \tag{35}$$

Here, $p(z)$ is the probability density function of a uniform distribution over a hypercube $[-\alpha, \alpha]^d$. It is defined as:

$$p(z) = \frac{1}{(2\alpha)^d} \cdot \mathbb{I}[z \in [-\alpha, \alpha]^d], \tag{36}$$

where $\mathbb{I}[z \in [-\alpha, \alpha]^d]$ is the indicator function, equal to 1 when all components of $z$ lie within the interval $[-\alpha, \alpha]$, and 0 otherwise. This ensures that the density is constant over the support and integrates to 1:

$$\int_{\mathbb{R}^d} p(z) \, dz = 1. \tag{37}$$

This convolutional smoothing can be understood as a low-pass filter on the loss surface: it averages the value of the loss over a local neighborhood, reducing sensitivity to high-frequency variations. As we will show in the next subsection, this results in an explicit curvature-penalizing regularization term.

### C.4 SECOND-ORDER EXPANSION

To analyze the regularization effect, we apply a second-order Taylor expansion of $L(\theta + z)$ around $\theta$:

$$L(\theta + z) \approx L(\theta) + \nabla L(\theta)^\top z + \frac{1}{2} z^\top HL(\theta)z, \tag{38}$$

where $HL(\theta) = \nabla^2 L(\theta)$ is the Hessian of the loss.

Taking the expectation over $z \sim \mathcal{U}(-\alpha, \alpha)^d$, we use the facts: - $\mathbb{E}[z_i] = 0$, - $\mathbb{E}[z_i z_j] = 0$ for $i \neq j$, - $\mathbb{E}[z_i^2] = \frac{\alpha^2}{3}$.

Then:

$$\mathbb{E}_z \left[ z^\top HL(\theta)z \right] = \sum_{i=1}^{d} H_{ii} \mathbb{E}[z_i^2] = \frac{\alpha^2}{3} \text{Tr}(HL(\theta)). \tag{39}$$

So the smoothed loss becomes:

$$\tilde{L}(\theta) = L(\theta) + \frac{\alpha^2}{6} \text{Tr}(HL(\theta)). \tag{40}$$

### C.5 INTERPRETATION

This shows that the perturbation introduces a curvature-dependent regularization term:

$$\tilde{L}(\theta) = L(\theta) + \frac{\alpha^2}{6} \sum_{i=1}^{d} \lambda_i, \tag{41}$$

where $\lambda_i$ are the eigenvalues of the Hessian. The trace term penalizes high curvature, effectively encouraging convergence to flatter regions of the loss landscape.

This regularization: (i) Damps sensitivity to small parameter shifts, (ii) Reduces susceptibility to sharp minima, (iii) Promotes better generalization.

Thus, RPO performs a *convolutional smoothing* of the policy gradient loss, without modifying the reward or requiring additional loss terms.

### C.6 SMOOTHING OF THE LOSS LANDSCAPE

A key effect of Robust Policy Optimization (RPO) is that it smooths the loss landscape by averaging the policy gradient objective over a local neighborhood in parameter space. This reduces the optimizer's sensitivity to sharp local variations and encourages convergence to flatter, more stable regions.

Formally, instead of minimizing the standard loss $L(\theta)$, RPO optimizes the smoothed objective:

$$\tilde{L}(\theta) = \mathbb{E}_{z \sim \mathcal{U}(-\alpha, \alpha)} \left[ L(\theta + z) \right], \tag{42}$$

which corresponds to convolving the original loss function with a uniform kernel. This operation averages the loss over a region around $\theta$, effectively dampening high-curvature fluctuations in the landscape.

Using a second-order Taylor expansion, we show (details in Appendix) that this smoothing introduces a regularization term proportional to the trace of the Hessian:

$$\tilde{L}(\theta) \approx L(\theta) + \frac{\alpha^2}{6} \text{Tr}(\nabla^2 L(\theta)). \tag{43}$$

This additional term penalizes sharp curvature in the loss function, encouraging updates that lead to flatter minima. As flatter solutions are known to generalize better in deep learning and reinforcement learning, this implicit smoothing mechanism contributes to RPO's improved robustness and stability without requiring any explicit regularization term or change to the reward structure.

While the loss landscape analysis above is derived under the assumption that perturbations are applied to all parameters, in **practice RPO perturbs only the action-related parameters**; we clarify this distinction below.

### C.7 ACTION-PARAMETER PERTURBATION IN PRACTICE

Although the above analysis is presented in terms of perturbing the full parameter vector $\theta$, in practice RPO applies perturbations only to the action-related parameters (e.g., the mean of the Gaussian policy) rather than the entire parameter set. The theoretical reasoning still holds in this restricted setting: the smoothing effect simply operates on a lower-dimensional subspace of $\theta$, and the corresponding Hessian-trace term reflects curvature only along those perturbed directions. In other words, RPO regularizes the optimization landscape with respect to the action parameters, which are most critical for policy improvement, while leaving other components unaffected. This preserves the intuition of implicit curvature penalization, but with the scope of smoothing constrained to the subspace of action parameters.

## D CHOICE OF NOISE DISTRIBUTION IN RPO

The choice of perturbation distribution plays a critical role in the effectiveness and stability of Robust Policy Optimization (RPO). In this section, we examine the theoretical properties and practical implications of using different noise distributions for parameter perturbation. Our focus is on three representative choices: uniform, Laplace, and Gumbel noise (Table 1).

Table 1: Properties of noise distributions considered for RPO perturbations

| Distribution | $\mathbb{E}[z]$ | $\mathrm{Var}(z)$ | Effect in RPO |
|---|---|---|---|
| Uniform $\mathcal{U}(-\alpha, \alpha)$ | 0 | $\frac{\alpha^2}{3}$ | Bounded, unbiased, analytically stable |
| Laplace $\mathrm{Lap}(0, b)$ | 0 | $2b^2$ | Heavy-tailed, promotes exploration but less stable |
| Gumbel $\mathrm{Gumbel}(\mu, \beta)$ | $\mu + \beta\gamma$ | $\frac{\pi^2\beta^2}{6}$ | Asymmetric, introduces gradient bias |

Laplace noise has a sharp peak and heavier tails than Gaussian, which can encourage exploration through larger and more frequent jumps in parameter space. However, this increased variance often leads to less stable learning and can hinder convergence in high-variance environments typical of reinforcement learning.

Gumbel noise, which is commonly used in sampling-based techniques, is asymmetric and has a nonzero mean $\mu + \beta\gamma$, where $\gamma \approx 0.577$ is the Euler–Mascheroni constant. This asymmetry introduces a directional bias in gradient updates, violating the zero-mean requirement needed for unbiased learning in RPO.

In contrast, uniform noise is symmetric, zero-mean, and bounded. These properties make it analytically tractable and ensure that perturbations remain within a well-controlled region of the parameter space. It enables closed-form derivations of the smoothing effect and regularization strength, while maintaining unbiasedness in gradient estimation.

We also provide an empirical comparison of these noise types in our ablation studies. While Laplace and Gumbel noise may encourage short-term exploration, they often result in higher performance variance and instability during training. In contrast, uniform noise consistently produces smoother learning curves and better generalization, confirming its suitability as the default perturbation model for RPO.

## E ADDITIONAL EXPERIMENTS DETAILS

**Implementation Details**: Our algorithm and the baselines are based on the PPO (Schulman et al., 2017) implementation available in (Huang et al., 2022a;b). This implementation incorporated many important advancements from existing literature in recent years on policy gradient (e.g., Orthogonal Initialization, GAE, Entropy Regularization). We refer reader to (Huang et al., 2022a) for further references. The experiments were conducted on a CPU-enabled machine, where each run of the algorithms generally took between 4 and 10 hours.

The pure data augmentation baseline RAD (Laskin et al., 2020) uses data processing before passing it to the agent, and the DRAC (Raileanu et al., 2020) uses data augmentation to regularize the loss of

Table 2: Hyperparameters for the experiments.

| Description | Value |
|---|---|
| Number of rollout steps | 2048 |
| Learning rate | $3e-4$ |
| Discount factor gamma | 0.99 |
| Lambda for the GAE | 0.95 |
| Number of mini-batches | 32 |
| Epochs to update the policy | 10 |
| Advantages normalization | True |
| Surrogate clipping coefficient | 0.2 |
| Clip value loss | Yes |
| Value loss coefficient | 0.5 |

value and policy network. We experimented with vector-based states and used *random amplitude scaling* proposed in RAD (Laskin et al., 2020) as a data augmentation method for RAD and DRAC. In the random amplitude scaling, the state values are multiplied with random values generated uniformly between a range $\alpha$ to $\beta$. We used the suggested (Laskin et al., 2020) and better performing range $\alpha = 0.6$ to $\beta = 1.2$ for all the experiments. Moreover, both RAD and DRAC use PPO as their base algorithm. However, our RPO method does not use any form of data augmentation.

We use the hyperparameters reported in the PPO implementation of continuous action spaces (Huang et al., 2022a;b), which incorporate best practices in the continuous control task. Furthermore, to mitigate the effect of hyperparameters choice, we keep them the same for all the environments. Further, we keep the same hyperparameters for all agents for a fair comparison. The common hyperparameters can be found in Table 2.

## F   ADDITIONAL RESULTS COMPARISON WITH PPO

**Results on All 48 tasks DeepMind Control** is in Figure 6. We observe that the performance improved in many of those environments and remain similar in others.

Results of *OpenAI Gym* environments: Pendulum and BipedalWalker are in Figure 7. Overall, our method, RPO, performs better compared to the PPO. In Pendulum environments, the PPO agent fails to learn any useful behavior in this setup. In contrast, RPO consistently learns with the increase in timestep and eventually learns the task. We see the policy entropy of RPO increases initially and eventually remains at a certain threshold, which might help the policy to stay exploratory and collect more data. In contrast, the PPO policy entropy decreases over time, and thus eventually, the performance remains the same, and the policy stops learning. This scenario might contribute to the bad performance of the PPO.

In the BipedalWalker environment, we see that both PPO and RPO learn up to a certain reward quickly. However, as we keep training both policies, we observe that after a certain period, the PPO's performance drops and even starts to become worse. In contrast, the RPO stays robust as we train for more and eventually keep improving the performance. These results show the robustness of our method when ample train time is available. The entropy plot shows a similar pattern, as PPO decreases entropy over time, and RPO keeps the entropy at a certain threshold.

Figure 8 shows results comparison on **IsaacGym** environments: Cartpole, and BallBalance. In this setup, we run the simulation up to 100M timesteps which take around 30 minutes for each run in each environment in a Quadro RTX 4000 GPU. We see that for Cartpole, both PPO and RPO learn the reward quickly, around 450. However, as we kept training for a long, the performance of PPO started to degrade over time, and the policy entropy kept decreasing. On the other hand, our RPO agent keeps improving the performance; notably, the performance never degrades over time. The policy entropy shows that the entropy remains at a threshold. This exploratory nature of RPO's policy might help keep learning and get better rewards. These results show the robustness of our method RPO over PPO even when an abundance of simulation is available. Interestingly, more simulation data might not always be good for RL agents. In our setup, the PPO even suffers from further training in

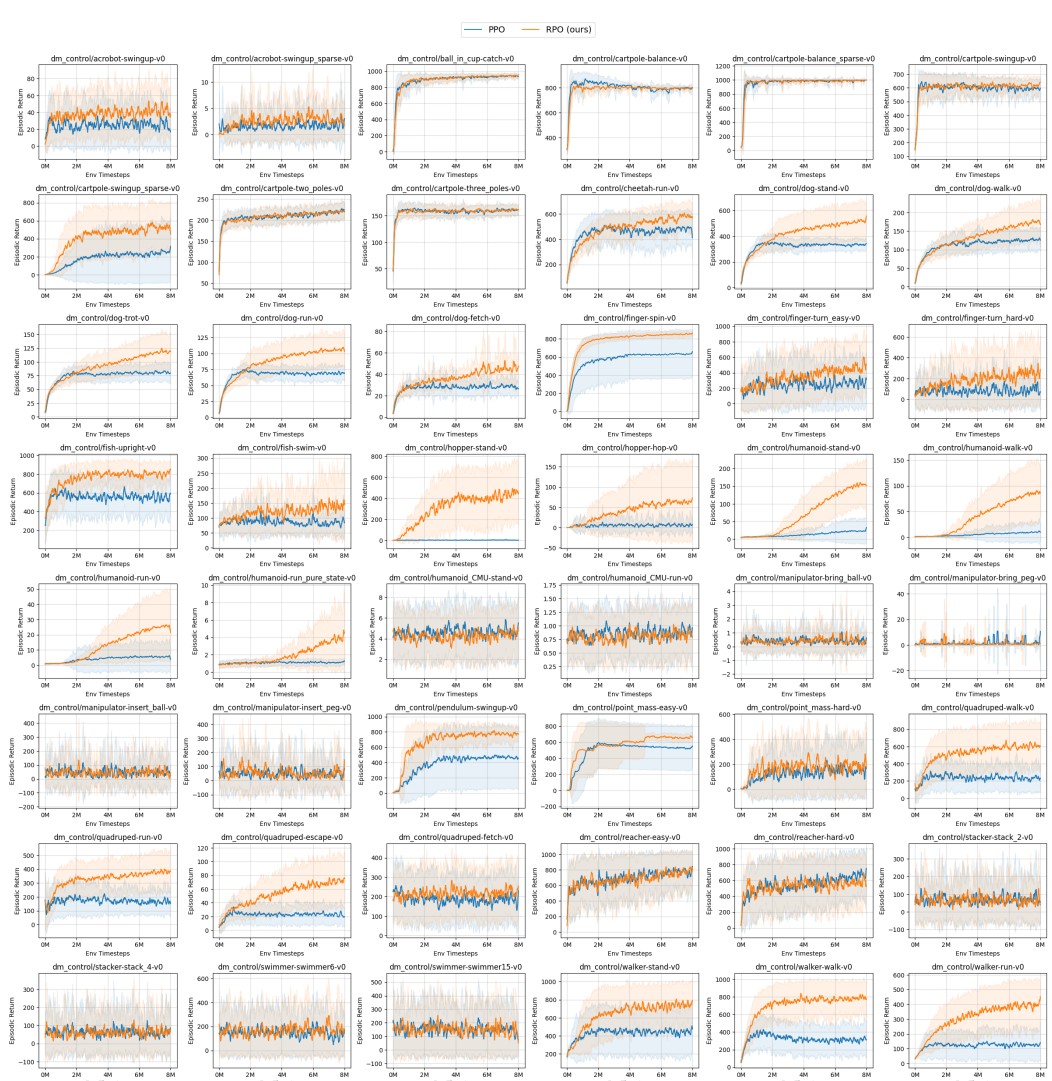

Figure 6: Our method RPO performs better or similarly in **48 DeepMind control environments**.

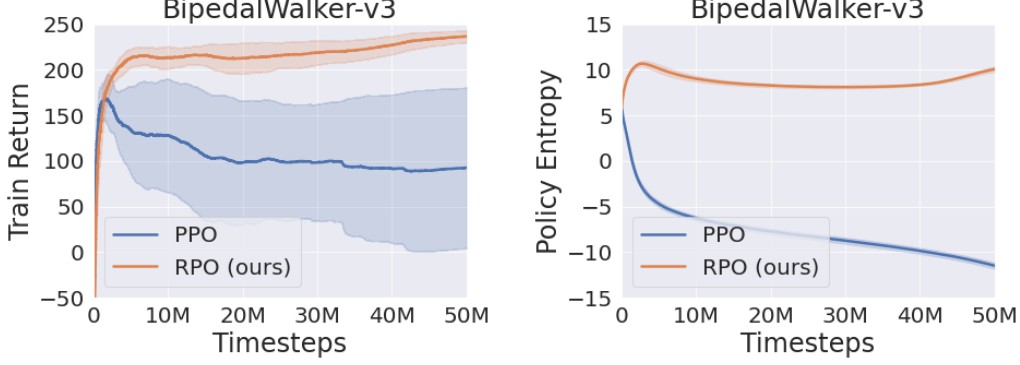

Figure 7: Results on OpenAI **Gym Environments**, BipedalWalker environment. Our RPO agent consistently improves its performance throughout training.

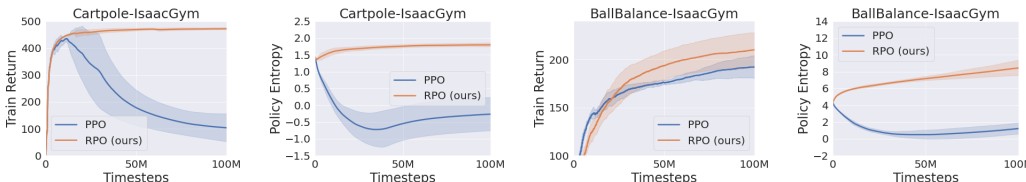

Figure 8: Results on **IsaacGym Environments**. In the Cartpole environment, the performance of PPO started to degrade over time, and the policy entropy kept decreasing. In contrast, our RPO agent keeps improving the performance over the entire training time. These results show the robustness of our method RPO over PPO even when an abundance of simulation is available. In the BallBalance environment, our method RPO achieves a slight performance improvement compared to PPO.

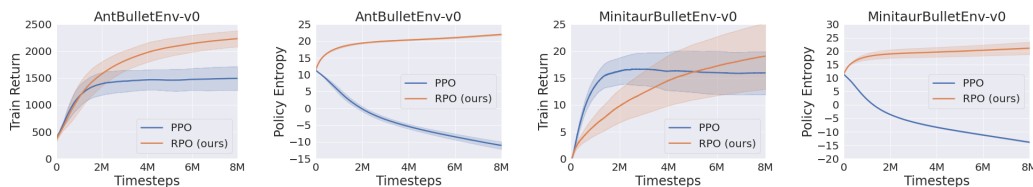

Figure 9: Results on **PyBullet Environments**. Our method RPO consistently performs better than the PPO in the Ant environment. On the other hand, in the Minitaur environment, PPO quickly learns up to a particular reward and remains on the same performance as time progresses while RPO surpasses the PPO's performance.

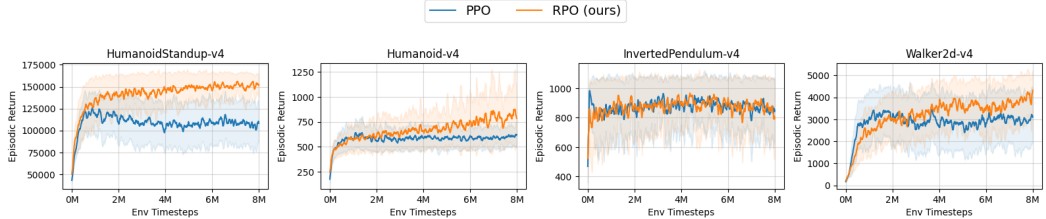

Figure 10: Results comparison of RPO with default $\alpha = 0.5$ on **Gym Mujoco-v4 environments**.

the Cartpole environment. In the BallBalance environment (results are averaged over 3 random seed runs), our method RPO achieves a slight performance improvement over PPO. Overall, our method RPO performs better than the PPO in the two IsaacGym environments.

Figure 9 shows results comparison on **PyBullet** environments: Ant, and Minituar. We observe that our method RPO performs better than the PPO in the Ant environment. In the Minitaur environment, PPO quickly (at around 2M) learns up to a certain reward and remains on the same performance as time progresses. In contrast, RPO starts from a lower performance, eventually surpassing the PPO's performance as time progresses. These results show the robustness of consistently improving the policy of the RPO method. The entropy pattern remains the same in both cases; PPO reduces entropy while RPO keeps the entropy at a certain threshold which it learns automatically in an environment.

Figure 10 shows the results comparison of RPO in **OpenAI Mujoco-v4** environments (with defaults $\alpha = 0.5$) values.

Additionally Figure 11 shows the result comparison with a tuned $\alpha = 0.01$ value. Overall, our method RPO improves of mathces the performance of PPO in these control tasks.

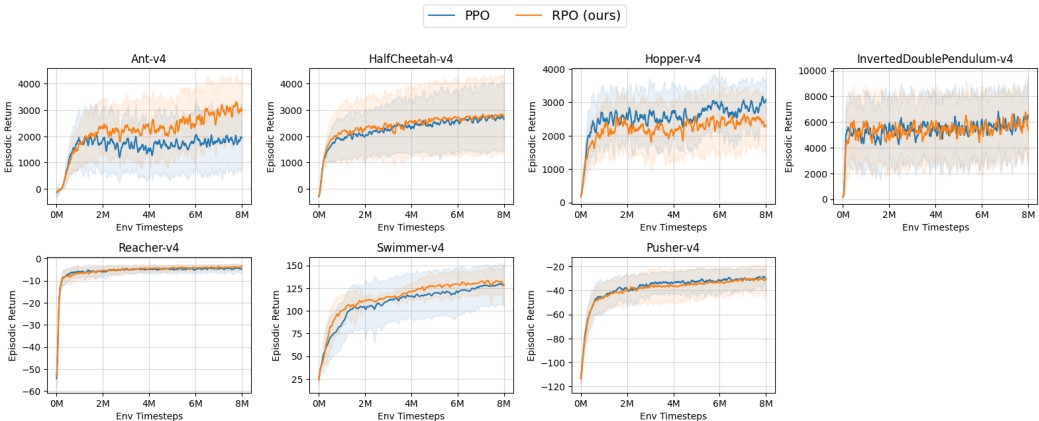

Figure 11: Results comparison of RPO with tuned $\alpha = 0.01$ on **Gym Mujoco-v4 environments**.

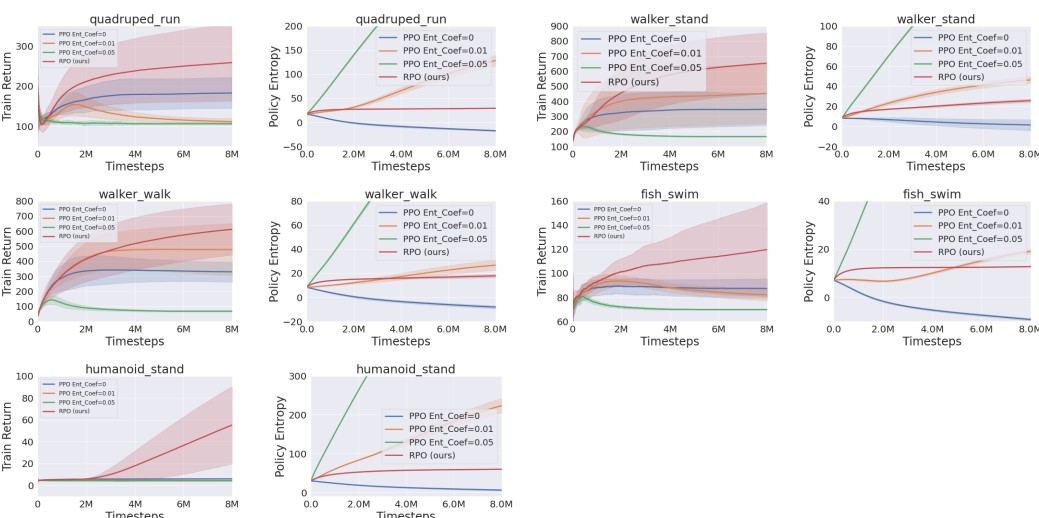

Figure 12: Results Comparison with **Entropy Regularization** on DeepMind Control.

# G   ADDITIONAL RESULTS COMPARISON WITH ENTROPY REGULARIZATION

The results comparison of RPO with entropy coefficient are in Figure 12. More entropy coefficient results are in Figure 13.

Figure 14 shows data augmentation return curve and entropy comparison with RPO.

# H   ABLATION STUDY - EFFECT OF $\alpha$

We conducted experiments on the $\alpha$ value ranges in the Uniform distribution. Figure 15 shows the return and policy comparison. We observe that the value of $\alpha$ affects the policy entropy and, thus, return performance. A smaller value of $\alpha$ (e.g., 0.001) seems to behave similarly to PPO, where policy entropy decreases over time, thus hampered performance. Higher entropy values, such as 1000.0, make the policy somewhat random as the uniform distribution dominates over the Gaussian distribution. This scenario keeps the entropy somewhat at a constant level; thus, the performance is hampered. Overall, a value between 0.1 to 3 often results in better performance. Due to overall performance advantage, in this paper, we report results with $\alpha = 0.5$ for all environments.

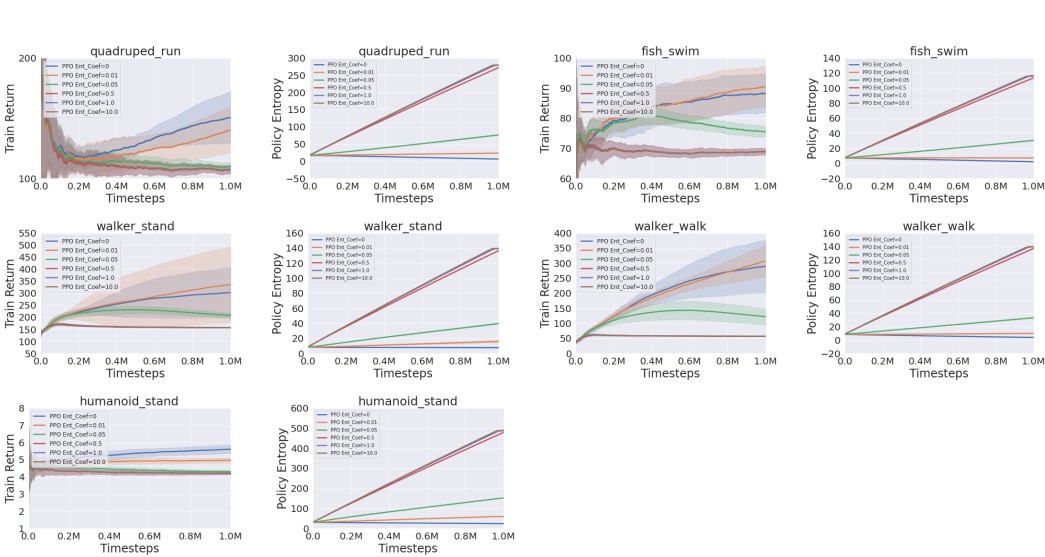

Figure 13: Comparison of return and policy entropy with **Entropy Regularization** on different (**more**) coefficient values in DeepMind Control Environments.

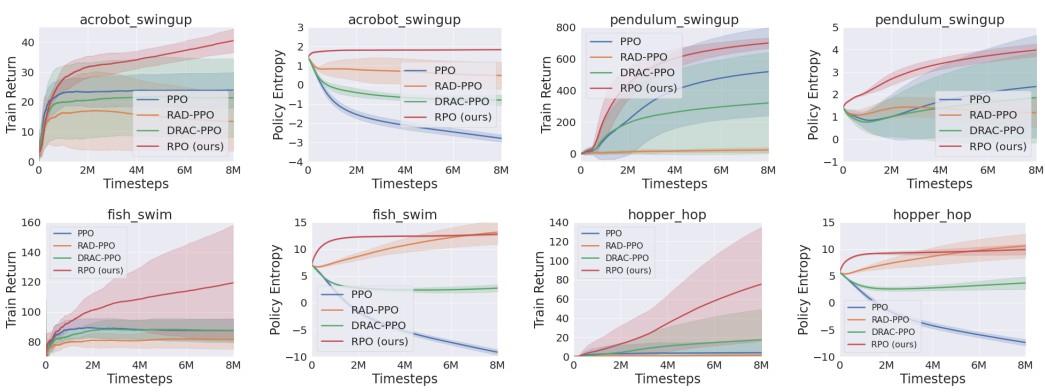

Figure 14: Comparison with PPO and **data augmentation** RAD, and DRAC on DeepMind Control.

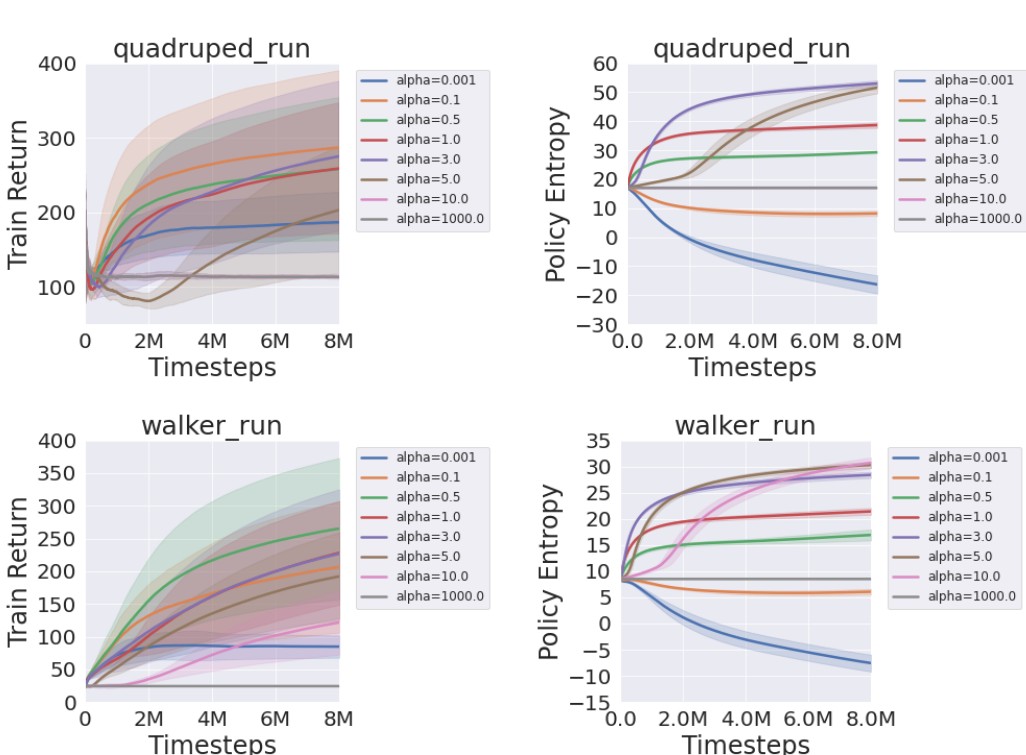

Figure 15: Ablation on $alpha$ values of the uniform distribution for RPO. An $\alpha$ value between $0.1$ to $3$ often results in better performance, while a large value often results in worse performance. The suggested default value is $\alpha = 0.5$.

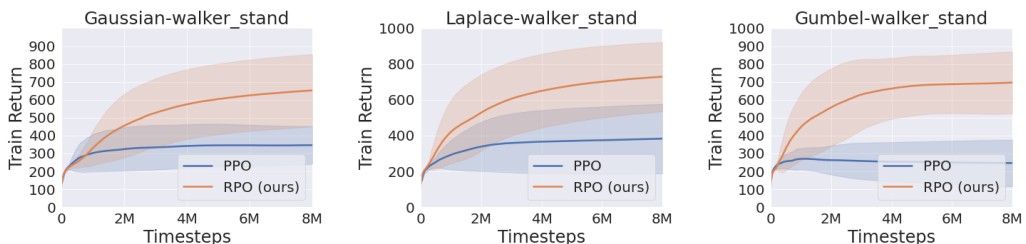

Figure 16: Results on different **action distributions**. Our method RPO shows improvement compared to base distributions. In these cases, the perturbed version of the distribution methods performs the best.

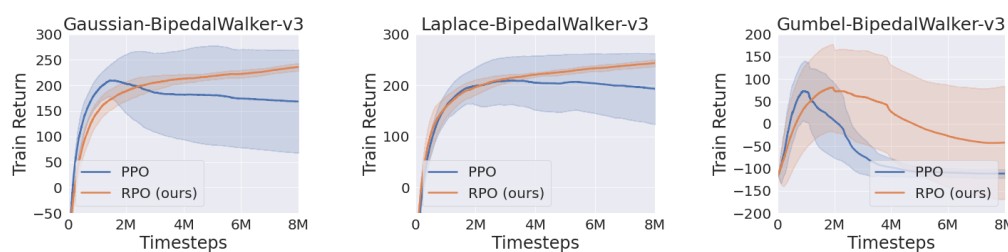

Figure 17: **Gym**: Results on different **action distributions**. Our method RPO shows improvement compared to base distributions.

## I    COMPARISON WITH OTHER (LAPLACE AND GUMBEL) DISTRIBUTIONS

In this section, we compare the results of the other two distributions: Laplace and Gumbel. We show empirical evidence that the choice of action distribution is important in solving a reinforcement learning task.

Furthermore, we combine our method of adding uniform distribution with the distributions (Gaussian, Laplace, and Gumbel). The results in Figure 16 show that our method overall improved the performance compared to the base distributions. These show the implication of our method in varieties of distributions. Our method also results in higher policy entropy throughout training, potentially improving exploration (see Figure 21, 22).

All Gym Environments results on different action distributions are in Figure 17.

Results on different action distribution on PyBullet environments are in Figure 18.

Results on different action distribution on DeepMind Control environments are in Figure 19.

## J    POLICY ENTROPY COMPARISON

Entropy Plot for DeepMind Control is in Figure 20.

Entropy Plot of Gym and Pybullet Environments for different action distributions is in Figure 21.

Entropy Plot of DeepMind Control for different action distributions is in Figure 22.

## K    ADDITIONAL RESULTS ON RPO-ALPHA ANNEALING

**Return:** Figure 23 shows a comparison of more results in other environments.

**Policy Entropy:** The policy entropy is impacted by the annealing of the $\alpha$ value, as depicted in Figure 24. The entropy initially increases and then begins to decline again as the training progresses

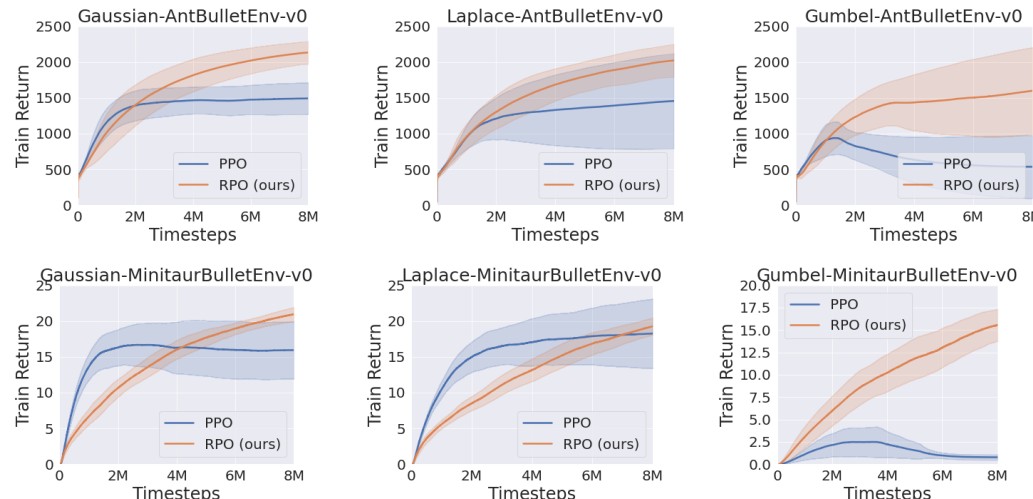

Figure 18: **PyBullet**: Results on different **action distributions**. Our method RPO shows improvement compared to base distributions.

towards its end. This indicates that the annealing approach provides more control over how the policy entropy changes during training.

## L    COMPARISON WITH DATA AUGMENTATION

We observe that the **data augmentation** method can help increase the policy's entropy by often randomly perturbing observations. This process might improve the performance where higher entropy is preferred. Thus, we compare our method with two data augmentation-based methods: RAD (Laskin et al., 2020), and DRAC (Raileanu et al., 2020). Here, The pure data augmentation baseline RAD uses data processing before passing it to the agent, and the DRAC uses data augmentation to regularize the value and policy network. Both of these data augmentation methods use PPO as their base RL policy.

Seeing the data augmentation through the lens of entropy, we observe that empirically, it can help the policy achieve a higher entropy than without data augmentation. However, this process often requires prior knowledge about the environments and a preprocessing step of the agent experience. Moreover, such methods might result in an uncontrolled increase in action entropy, eventually hampering the return performance (Raileanu et al., 2020). The results on DeepMind Control environments are shown in Table 3. Our method performs better in mean episodic return in most environments than PPO and other data augmentation baselines RAD and DRAC.

We observe that the data augmentation slightly improved the base PPO algorithms, and the policy entropy shows higher than the base PPO. However, our method RPO still maintains a better mean return than all the baselines. The entropy of our method shows an increase at the initial timestep of the training. However, it eventually becomes stable at a particular value. The data augmentation method, especially RAD, shows an increase in entropy throughout the training process. However, this increase does not translate to the return performance. Moreover, improper handling of the data augmentation may result in worsen performance.

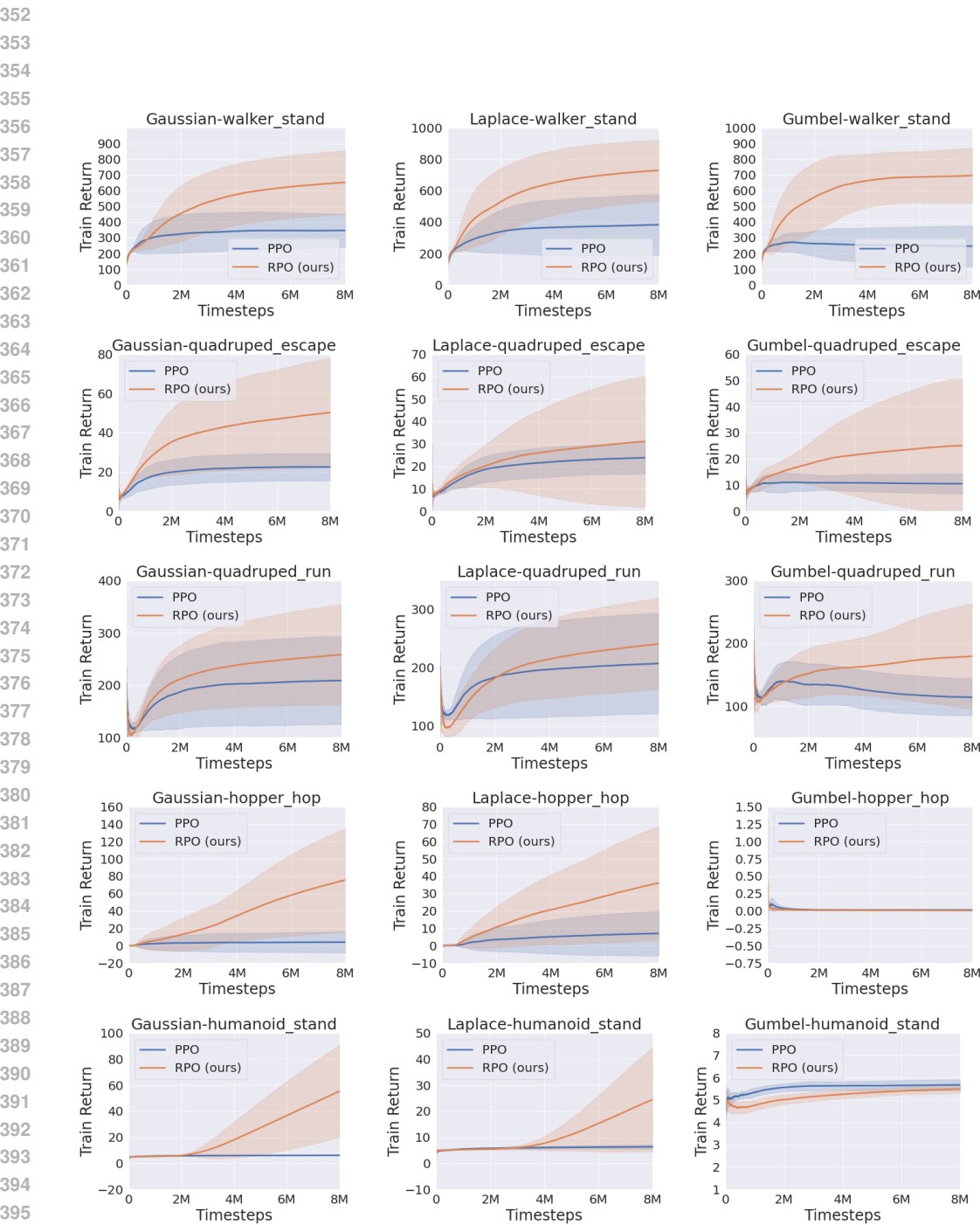

Figure 19: DeepMind Control: Results on different **action distributions**. Our method RPO shows improvement compared to base distributions in many environments.

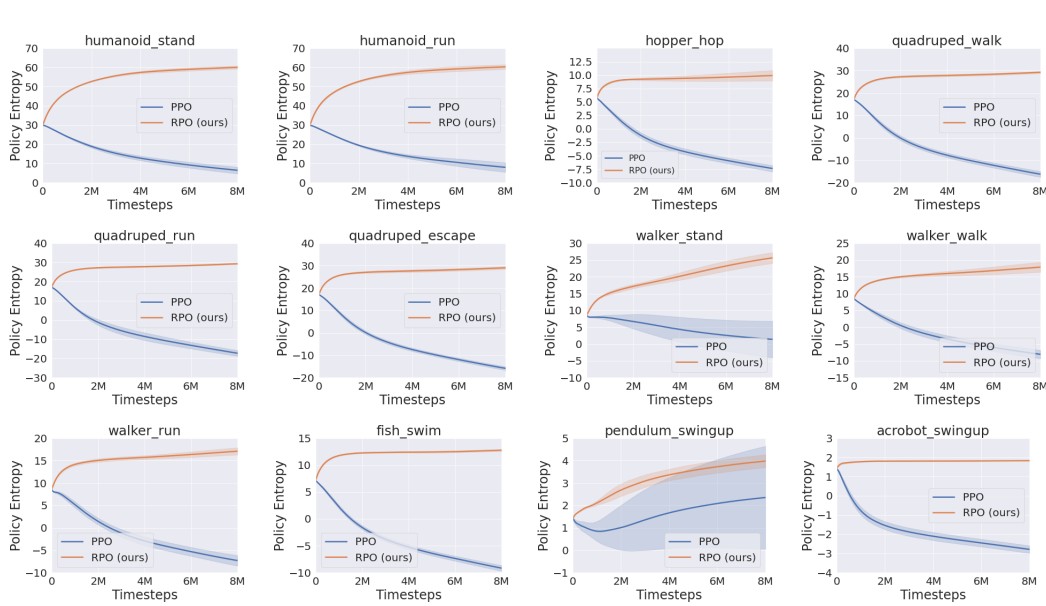

Figure 20: **Entropy** Results on **DeepMind Control Environments**.

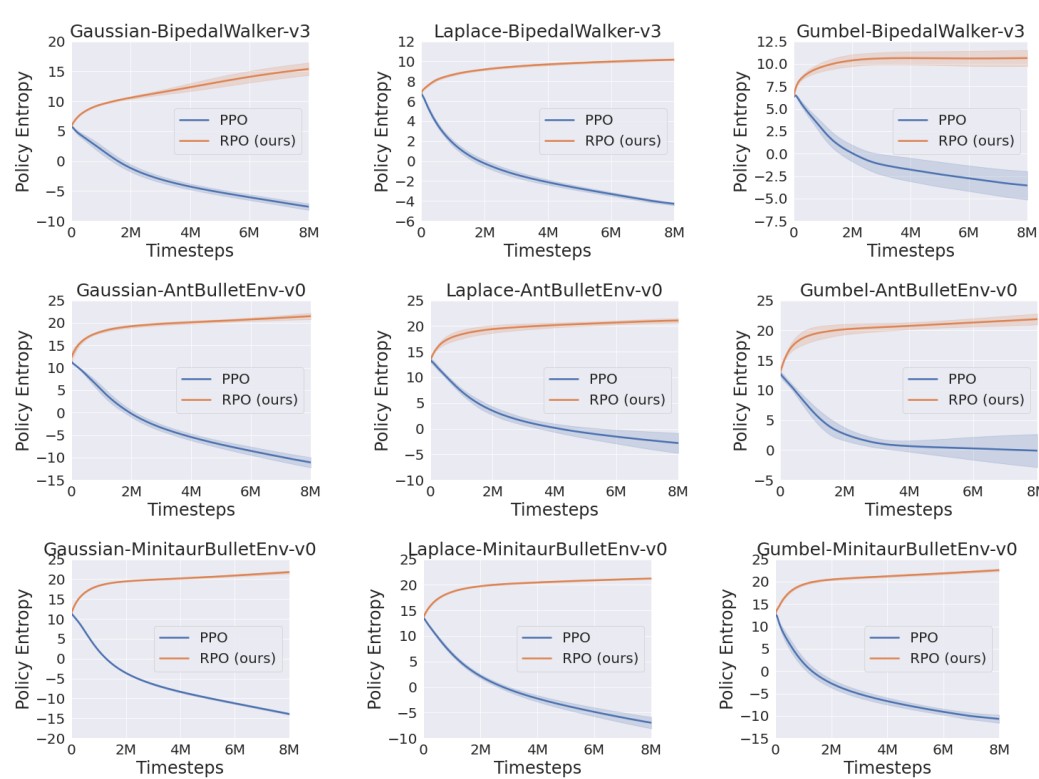

Figure 21: **Gym and Pybullet: Entropy** comparison on different action distribution. In all cases, our method of perturbing distribution with Uniform distribution (RPO) results in higher policy entropy and potentially improved exploration.

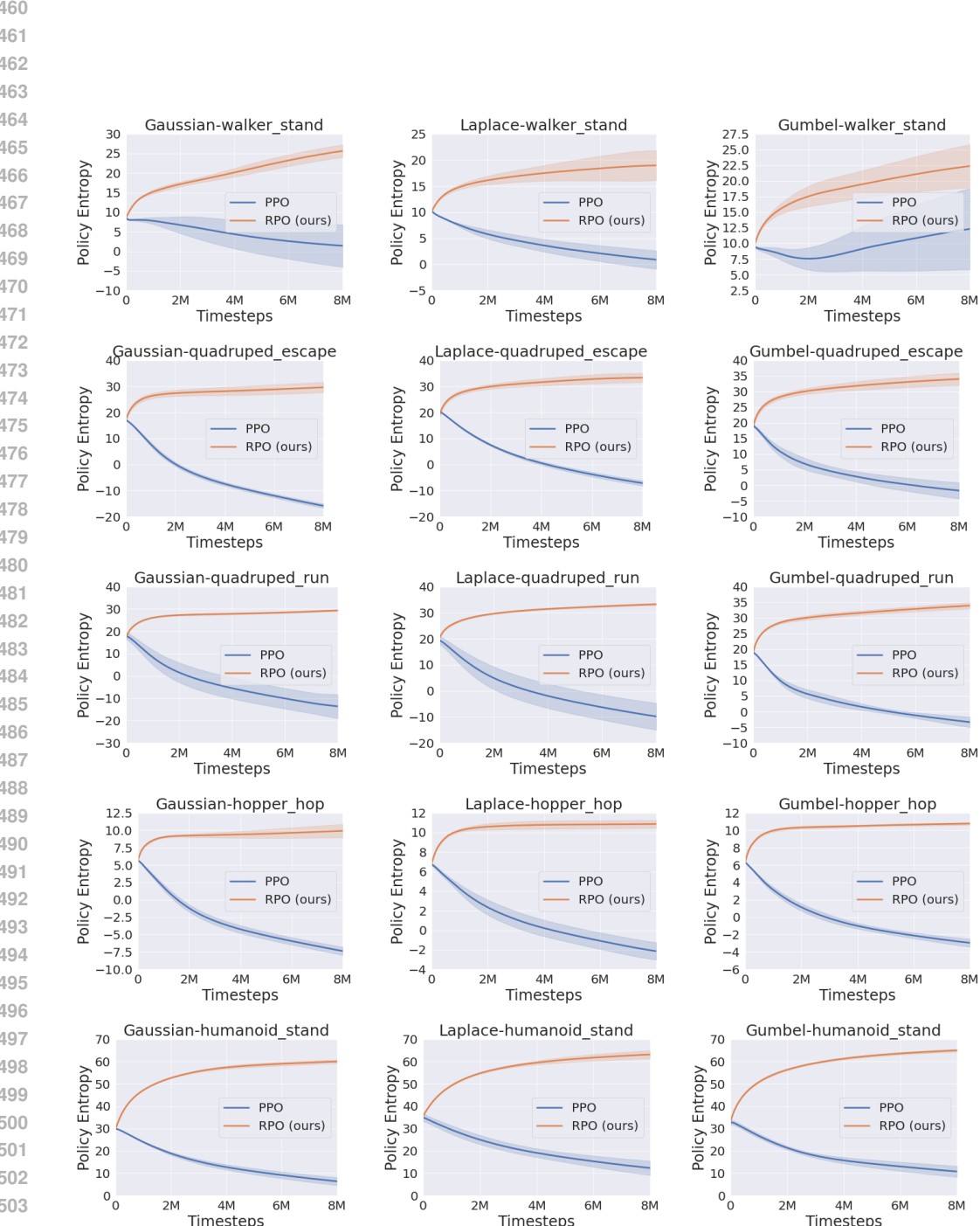

Figure 22: **DeepMind Control**: Policy **Entropy** Results on different **action distribution**.

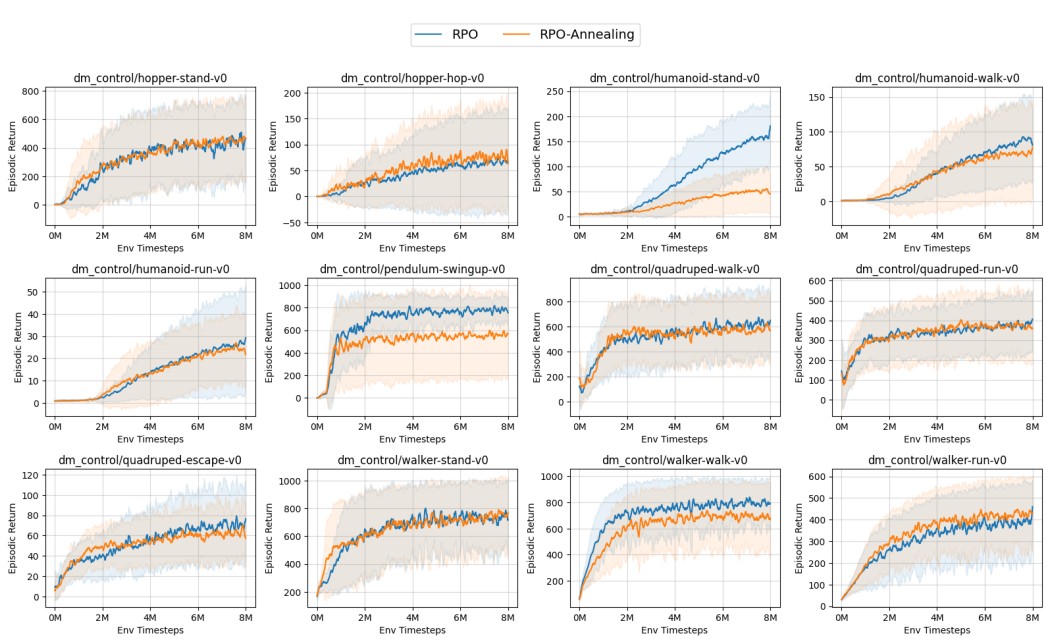

Figure 23: RPO vs **RPO-Annealing Return** Comparison.

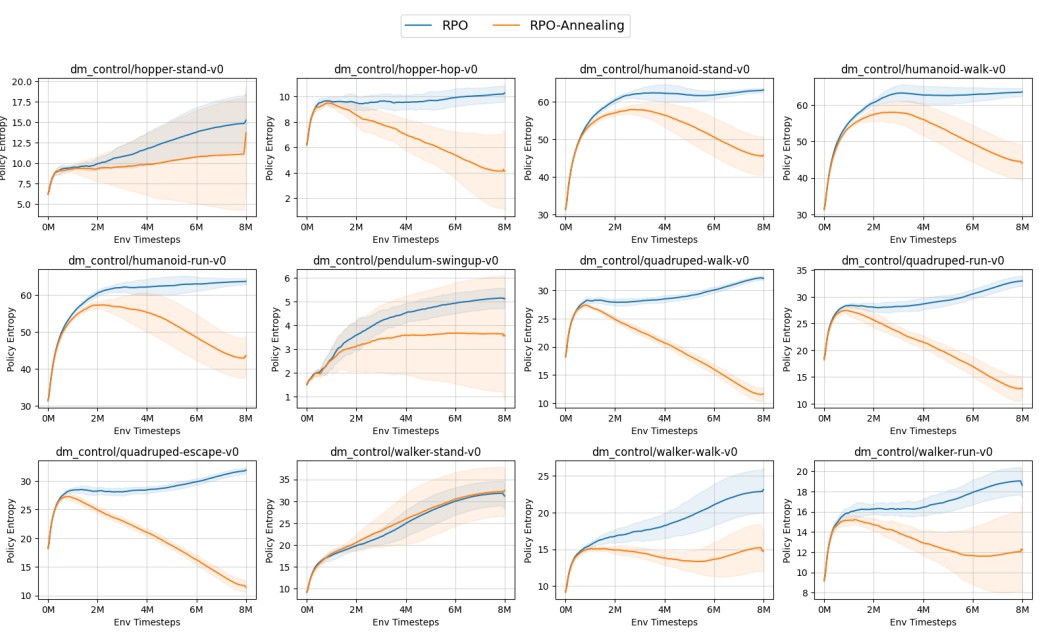

Figure 24: RPO vs **RPO-Annealing Entropy** Comparison. The entropy initially rises and then falls as training progresses, suggesting that the annealing approach offers greater control over policy entropy changes during training.

Table 3: Result comparison with **data augmentation on DeepMind control environments**. Our method RPO performs better in mean episodic return in most environments than PPO and other data augmentation baselines RAD and DRAC. Moreover, in some environments, the data augmentation baselines worsen the performance compared to the base PPO. The results are after training the agent for 8M timesteps. The mean and standard deviations are over 10 seed runs.

| Env | PPO | RAD-PPO | DRAC-PPO | RPO (ours) |
|---|---|---|---|---|
| acrobot swingup | 23.93 ±6.0 | 13.41 ±9.84 | 21.32 ±13.17 | **40.46** ±4.01 |
| fish swim | 87.39 ±8.0 | 81.53 ±6.53 | 87.5 ±7.74 | **119.42** ±38.46 |
| humanoid stand | 6.06 ±0.17 | 6.14 ±0.37 | 37.58 ±66.09 | **55.22** ±35.34 |
| humanoid run | 1.55 ±1.59 | 1.1 ±0.04 | 13.13 ±16.6 | **13.26** ±9.45 |
| pendulum swingup | 518.65 ±279.73 | 23.04 ±17.08 | 320.51 ±319.76 | **699.79** ±32.97 |
| quadruped walk | 216.67 ±66.56 | 395.07 ±124.91 | 374.78 ±197.98 | **437.66** ±191.93 |
| quadruped run | 183.14 ±38.48 | 158.01 ±26.84 | 247.02 ±83.23 | **258.75** ±96.18 |
| quadruped escape | 22.51 ±7.0 | **54.0** ±22.13 | 52.62 ±21.89 | 53.37 ±25.73 |
| walker stand | 346.55 ±104.87 | 558.92 ±131.09 | 361.99 ±93.44 | **652.7** ±202.47 |
| walker walk | 329.61 ±67.92 | 333.22 ±84.38 | 425.51 ±150.52 | **611.88** ±170.46 |
| walker run | 123.63 ±53.5 | 141.43 ±34.77 | 132.28 ±37.18 | **291.83** ±108.8 |

