# OpenReview forum: "Robust Policy Gradient Optimization through Action Parameter Perturbation in Reinforcement Learning"
_ICLR.cc/2026/Conference — Submitted to ICLR 2026_

### Official Review · Reviewer_2reW · 2025-10-24

**Soundness:** 2
**Presentation:** 3
**Contribution:** 2
**Rating:** 4
**Confidence:** 4

**Summary:**

This paper proposes the robust policy optimization (RPO) algorithm that introduces policy parameter perturbations during optimization and smooths the loss landscape and enhances regularity and stability.

**Strengths:**

Overall a clear paper with easy-to-follow presentation. Extensive experiments are conducted to compare the proposed RPO algorithm with the PPO baseline. In fact, the overall performance of PPO is poor on the DM Control benchmark, compared to off-policy methods (see https://arxiv.org/pdf/2501.16142?, Table 5). Therefore, it is interesting to see the significant improvement by RPO.

**Weaknesses:**

+ **Unconvincing theoretical analysis.**

In Section 4 and Appendix C, it states that RPO smooths the optimization landscape by perturbing the action parameter, which is feasible because the uniform sampling serves as a convolution applied to the original objective. While it is true that convolving to another smooth function grants the same level of smoothness, the probability density function of a uniform distribution is not smooth everywhere as it has discontinuities at the boundary.

Moreover, the current presentation focuses on how the proposed algorithm smooths the objective vis action perturbations. However, it remains unclear why smoothing the landscape is necessary and beneficial for the policy training (e.g., giving illustrations on how non-smooth the original objective can be). The motivation presented in the paper is too vague to let the audience fully understand the significance of the proposed method.

+ **Incomplete experimental details.**

It seems that the implementation details of RPO are missing, only PPO hyperparameters are provided. Since the core contribution of this work is the perturbation method that smooths the objective, it is necessary to clarify if any extra techniques, such as ratio clipping and mini-batching, are applied to the RPO implementation.

It is well-known that the implementation of PPO relies on tons of tricks, which makes it hard to isolate and identify the sole effect of the perturbation method. For example, the PPO objective clips those probability ratios which exceed certain threshold, making the actual gradient deviated from the true gradient. This bias persists in RPO if it is built on the PPO pipeline, which may cause discrepancy between the theory and the practice, and should be elaborated and analyzed in detail.

**Questions:**

Please see Weaknesses.

---

### Official Review · Reviewer_eb67 · 2025-10-27

**Soundness:** 1
**Presentation:** 2
**Contribution:** 1
**Rating:** 0
**Confidence:** 4

**Summary:**

This paper proposed to inject uniform noises into the mean of Gaussian policy so that in optimization the score function is perturbed in a hope to bring more stable optimization and better performance.

**Strengths:**

The paper nicely points out that gradient based policy optimization is local in nature, prone to overfit to short-horizon rewards. Section 3 comprehensively surveys existing works on noise injection from action/parameter-level noise for exploration to data augmentation. Overall the proposed method is simple and understandable.

**Weaknesses:**

There are many reasons why this paper is not ready for publication. But perhaps the first is its unclear motivation. The authors name the proposed method as RPO, but it remains unclear to me robust against what. In the experiments RPO is compared against standard baselines on standard environments, there is no mention of the target: noise? adversary environment? or numerical error that RPO is up against. Higher return is better, but there is no explanation why and how the designed robustness is responsible for that improvement.
From a technical perspective, the analysis is too simple if not trivial. I believe for the paper to be a nontrivial contribution to the community it is expected that convergence rate be established given many prior works on PPO/TRPO. For example the authors could show how much convergence rate is sacrificed in exchange for what level of robustness.

**Questions:**

please see the weaknesses section

---

### Official Review · Reviewer_tV1w · 2025-10-29

**Soundness:** 2
**Presentation:** 3
**Contribution:** 2
**Rating:** 4
**Confidence:** 4

**Summary:**

The paper proposes a policy gradient algorithm that perturbs the mean of the policy in the loss. This perturbation is added to increase the robustness of the optimization by discouraging small policy variance, and is presented as an alternative to the entropy bonus in algorithms like PPO. The authors evaluate the method on popular simulated RL benchmark tasks.

**Strengths:**

The modifications to the base policy gradient algorithm are conceptually simple and should be easy to add to existing implementations. The evaluation is done on a large number of tasks, and on most tasks, the performance is either on par with or better than that of PPO.

**Weaknesses:**

1. To me, it is not clear where the performance gains of the method compared to PPO (with entropy bonus) are coming from. The authors show in equation 13 that the proposed objective is equivalent to adding a penalty for small policy variances. The entropy term in PPO likewise penalizes small policy variances, and it is unclear to me why one would work better than the other.

2. Figure 2 only shows results for three entropy coefficient values. Both ent_coef=0.01 and ent_coef=0.05 result in policy entropies that are significantly larger than the policy entropy of RPO. It is therefore unclear whether the performance difference between PPO and RPO is due to the different entropy handling or whether the tested entropy coefficients are simply too large. More runs with ent_coef < 0.01 would help clarify this point.

3. The paper repeatedly states that one advantage of the method is that it does not have an entropy coefficient that needs to be tuned, e.g., "RPO removes the need for tuning entropy coefficients". However, the method introduces a new hyperparameter $\alpha$, which, according to Figure 3, also has a decent impact on the policy performance. I would appreciate it if the authors could elaborate on why it is better to have this hyperparameter compared to the entropy coefficient hyperparameter.

**Questions:**

1. The idea of this paper seems very related to adding noise to the gradient, which has led to performance gains in the supervised learning literature [1]. Would adding noise to the gradient have a similar effect here, or is there an advantage of adding the noise to the output?

2. Figure 3 shows that the policy entropy essentially stagnates at the initial value for $\alpha=1000$. This is a bit puzzling to me since in this case, the perturbations are significantly larger than the range of actions of the environment. With such strong perturbations, I would expect that the optimizer can only decrease the loss by increasing the policy variance and therefore the entropy, but this does not seem to happen here. Perhaps the authors could give an intuition why the entropy is not increasing here.

3. The paper often talks about "perturbations to the policy parameters" and the derivations in Appendix C also assume general perturbations to the network parameters, yet all the experiments only perturb the output of the network. Is there a reason for this? How would the performance of RPO change if the method perturbed the parameters of the network instead of just the output?

4. How was the $\alpha$ chosen in Figures 1, 6-9? Do all experiments use the same value, or is $\alpha$ tuned for each task? More generally, are the hyperparameters for RPO and PPO tuned for each task, or do all experiments use the same hyperparameters?

5. Equation 13: The text states that this relation results from a Taylor approximation, but the derivation in Appendix B does not seem to use a Taylor expansion. I would appreciate it if the authors could clarify where the Taylor expansion comes into play here.


Comments:
1. In multiple locations, the paper simply references the appendix without specifying the section in the appendix. This makes it a bit tedious to find the content that is referenced.

[1] Neelakantan, Arvind, et al. "Adding gradient noise improves learning for very deep networks." arXiv preprint arXiv:1511.06807 (2015).

---

### Official Review · Reviewer_vNJs · 2025-11-01

**Soundness:** 2
**Presentation:** 3
**Contribution:** 2
**Rating:** 4
**Confidence:** 3

**Summary:**

This paper proposes Robust Policy Optimization (RPO), an on-policy policy gradient method that introduces perturbations to the action parameters (specifically the mean of a Gaussian policy) during the optimization phase, while leaving the data collection process unchanged. The authors argue that this approach smooths the loss landscape, implicitly regularizes the optimization, and biases updates toward flatter regions, leading to improved convergence, higher returns, and greater robustness compared to baselines like PPO and entropy-regularized PPO. Experiments on various continuous control tasks from benchmarks demonstrate superior performance in most of the environments.

**Strengths:**

- The approach is simple, computationally lightweight, and easily applicable to existing PPO-style methods.
- Extensive experiments consistently show improvements in learning stability and final return.
- Provides a link between parameter-space perturbation and loss-smoothing through a second-order (Hessian) approximation.

**Weaknesses:**

- Missing Citation and Comparison to Key Prior Work

The paper's novelty is highly questionable due to the omission of the most relevant prior work: "Flat Reward in Policy Parameter Space Imples Robust Reinforcement Learning" (Lee & Yoon, ICLR 2025). That paper also identifies the connection between "flatter reward landscapes" and "robustness" in RL and proposes an optimization-time solution (SAM+PPO). The core conceptual motivation of RPO is therefore not new. A thorough discussion of, and comparison to, SAM+PPO is completely missing and is a major flaw.

- Limited theoretical contribution.

The analytical results mainly restate the unbiasedness of the estimator and a local Hessian penalty approximation.
No convergence or robustness theorem (beyond optimization flatness) is provided, and the analysis remains heuristic.

**Questions:**

- How does RPO differ mechanistically from the flat reward analysis in Lee & Yoon (2025)? Specifically, does perturbing only the action mean (vs. full parameters) limit the smoothing effect compared to broader parameter perturbations?

- Can the authors provide a more formal justification for the gap between the theory (perturbing $\theta$) and the practice (perturbing $\mu$)? Does perturbing $\mu$ truly penalize the trace of the full policy's Hessian, or only a very specific sub-block related to the action head?

---

### Meta-Review · Area_Chair_S53W · 2026-01-03

**Summary:**

Here is a summary of the reviewers' concerns:
- Limited novelty: The core conceptual motivation of RPO is not new, and is highly related to adding noise to the gradient.
- Limited experiments: Figure 2 only shows results for three entropy coefficient values. missing some baselines.
- Limited theoretical justification and contribution: No convergence or robustness theorem (beyond optimization flatness) is provided, and the analysis remains heuristic.
- Missing ablation study to identify the source of improvement.
- Lack deep analysis: it is nor clear why the introduced new hyperparameter \alpha is better than the entropy coefficient hyperparameter.
- Missing Citation and Comparison to Key Prior Work: it is unclear how the proposed method differ from adding noise to the gradient. A thorough discussion of, and comparison to, SAM+PPO is also missing.
- Missing technical details: the implementation details of RPO are missing; it is unclear how the hyperparameters for RPO and PPO are tuned for each task, and whether all experiments use the same hyperparameters.

**Reviewer Concerns:**

The authors did not make a rebuttal. All reviewers' concerns remain.

**Reviewer Scores:**

All reviewers are likely to keep their scores since there is no rebuttal.

---

### Decision · Program_Chairs · 2026-01-26

Reject